# TARE: Lightweight Token-Aware Representation Editing for Fine-tuning Transformer

## Abstract

Parameter-efficient fine-tuning (PEFT) of large Transformers often struggles to balance effectiveness with efficiency. Methods based on low-rank adaptation can be resource-intensive, while representation-editing techniques that apply a single, global transformation tend to underfit fine-grained, token-level contexts. The core challenge is achieving token-aware, fine-grained edits while keeping inference overhead and the hyperparameter tuning burden negligible. Our work introduce Token-Aware Representation Editing (TARE), a novel PEFT method. After each feed-forward network (FFN) block, TARE employs a lightweight selector that scores a small pool of "editors" for each token's hidden representation. It sparsely activates only the top-scoring editors and mixes their element-wise edits to update the representation. Because the edits are computationally minimal diagonal operations and are sparsely activated, TARE adds near-zero inference overhead and introduces no rank or scaling hyperparameters. Our work conduct extensive experiments on LLaMA-3-8B across eight knowledge reasoning and seven mathematical reasoning tasks, and on RoBERTa-base/large for the GLUE benchmark. Compared to strong baselines like LoRA, DoRA, MiLoRA, LoReFT, and RED, TARE achieves state-of-the-art results. It attains an 86.7% average on knowledge reasoning tasks, 76.7% on mathematical reasoning tasks, and 88.3% on the GLUE benchmark. These results are achieved while tuning only 0.0392% of the model's parameters and using approximately 20 GiB of memory, surpassing prior methods by several percentage points and demonstrating exceptional resource efficiency. An anonymized implementation is available at: https://anonymous.4open.science/r/tare-BCF5/.

## 1 Introduction

Parameter–efficient fine–tuning (PEFT) has become a central paradigm for adapting large Transformers under tight compute and memory budgets: it aims to reach strong task performance by training only a tiny fraction of parameters while keeping the backbone frozen. Existing PEFT families include weight–space adapters (e.g., LoRA Hu et al. (2021), DoRA Liu et al. (2024), MiLoRA Wang et al. (2024a)), representation–space editing and gating (e.g., RED Wu et al. (2024a), LoReFT Wu et al. (2024b), IA$^3$ Liu et al. (2022), BitFit Ben Zaken et al. (2021)). Despite clear efficiency gains, a key open problem remains: how to attain fine-grained, token-aware adaptation while keeping inference overhead and hyperparameter burden negligible.

Across methods, a common limitation is the tension between expressiveness and efficiency. Low-rank approaches such as LoRA Hu et al. (2021), DoRA Liu et al. (2024), and MiLoRA Wang et al. (2024a) require choosing ranks and scaling factors, which can complicate tuning across layers and tasks. Importantly, in standard single-adapter deployments these low-rank increments are merged into the base weights, so there is effectively no additional inference overhead. When weight merging is not desirable—e.g., multi-adapter hot-swap, online mixture/selection, or coexistence with certain quantization pipelines—one may resort to on-the-fly composition, which re-introduces extra operators, but this is an engineering choice rather than an inherent property of LoRA-style methods. Representation-editing methods that are highly efficient at inference often apply a single, shared transformation to all tokens—e.g., RED Wu et al. (2024a) learns one global per-feature scaling/bias; IA$^3$ Liu et al. (2022) gates channels uniformly; BitFit Ben Zaken et al. (2021) updates only biases—thereby limiting capacity to capture fine-grained context. LoReFT Wu et al. (2024b) performs

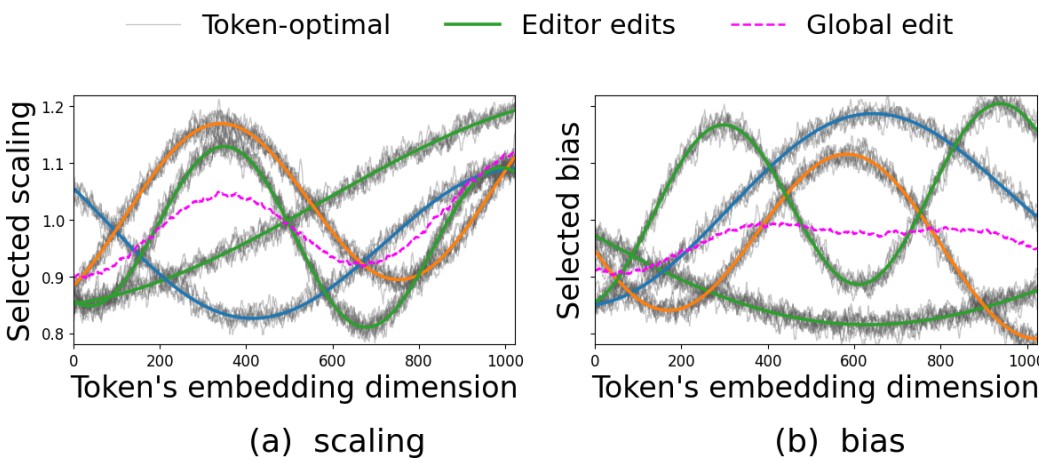

Figure 1: Token-wise optimal scaling/bias (Token-optimal) forms a few modes. A single global scaling/bias (Global edit) underfits, a small set of scalings/bias (Editor edits) covers dominant modes.

low-rank projections in representation space but still uses the same projection for every token and inherits rank-selection overhead. In summary, many methods either impose a uniform editor that underfits token-level variability, or they rely on hyperparameters (e.g., ranks/scales) and deployment choices (e.g., merging vs. online composition), motivating a token-aware representation editor that preserves inference efficiency while capturing per-token context.

As shown in Figure 1, for a single layer, the per-dimension scaling (top) and bias (bottom) that would be individually optimal for different tokens (thin solid curves). Two regularities emerge. First, token requirements are highly heterogeneous across embedding dimensions: the thin curves span roughly 0.8–1.2 for both scaling and bias and exhibit clear phase shifts, indicating that different tokens prefer amplifying/suppressing different feature bands. Second, despite this heterogeneity, the thin curves concentrate around a small number of prototypical shapes (thick solid curves); most token-specific curves closely follow one of these smooth templates up to modest perturbations. In contrast, the single global edit (thin dashed) is essentially the per-dimension average; it flattens peaks and valleys and therefore underfits wherever tokens require opposite adjustments (e.g., around the mid- and high-dimensional regions where one mode rises while another falls). The same multi-modal pattern appears simultaneously in both scaling and bias, and the two often exhibit slight phase misalignment, suggesting that accurate edits must coordinate the pair rather than rely on either alone. This analysis implies that token-level edits are necessary to capture fine-grained semantics, and only a few hidden representation editors are sufficient to cover the dominant modes.

Consequently, our work proposed **Token-Aware Representation Editing** (*TARE*), which adopts a token-aware hidden representation editing scheme. TARE inserts a hidden representation editor module after each block's FFN: for each token, a lightweight selector produces logits over $n$ diagonal editors and activates only the Top-$k$ hidden representation editors; each selected hidden representation editor maintains element-wise scale and bias vectors $(\gamma_i, b_i)$ to form cansdidate edits $h_i = h_1 \odot \gamma_i + b_i$, which are then linearly mixed by softmax-normalized weights to update the representation. Because the operations are diagonal along feature dimensions and selection is sparse, the inference overhead is nearly unchanged; the backbone network of large Transformer is frozen, and only $(n, k)$ need to be set—no rank/scale hyperparameters are introduced.

The main contributions of this work are as follows:

- Our work propose **Token-Aware Representation Editing** (*TARE*), a new PEFT mechanism that replaces one-size-fits-all edits with per-token, per-dimension adjustments. A lightweight selector scores a small pool of hidden representation editors and mixes only a few of them for each token, yielding fine-grained context adaptivity while keeping computation strictly diagonal and sparse. This directly tackles the key challenge raised

above—achieving token-level expressiveness without adding inference latency or complex hyperparameters.

- Our work show that token-optimal edits cluster into a handful of smooth modes; the proposed TARE method's selector–template co-design exploits this structure by projecting each token onto a local convex combination of learned hidden representation editors. This design preserves the inference friendliness of representation editing, avoids rank/scale knobs from low-rank adapters, and provides a simple, robust training recipe with optional load-balancing regularization.

- The proposed TARE method is evaluated on a decoder model (LLaMA-3-8B) across eight knowledge reasoning tasks and seven mathematical reasoning tasks, and on encoder models (RoBERTa-base/large) on GLUE benchmark. It achieves 86.7% average over eight knowledge reasoning tasks (slightly above LoReFT and notably higher than LoRA/RED), 76.7% average over seven mathematical reasoning tasks and 88.3% on GLUE benchmark, while tuning only 0.0392% of parameters with ∼20 GiB memory. TARE consistently matches or surpasses strong PEFT baselines (LoRA, DoRA, MILoRA, RED, LoReFT) under tight parameter and memory budgets.

## 2 RELATED WORK (A.2)

## 3 TOKEN-AWARE REPRESENTATION EDITING

This section introduces the proposed TARE method. Rather than using dense low-rank adapters, TARE employs a lightweight, token-wise selector. For each token, it activates a small set of hidden representation editors (per-feature scaling and bias) and mixes their edits with normalized weights. This token-aware, $k$-sparse, diagonal adjustment increases expressiveness and captures fine-grained context. It adds virtually no inference overhead and avoids rank/scale hyperparameters. As a result, TARE transfers well across diverse tasks while alleviating the extra computation and overfitting issues of conventional fine-tuning.

### 3.1 DESIGN PRINCIPLES

**Notation and setup.** Fix a Transformer layer index $\ell$. Let $h_{\ell,t} \in \mathbb{R}^{1 \times 1 \times D_\ell}$ denote the hidden representation of a given token $t$ at layer $\ell$. A diagonal hidden representation editor applies a feature-wise affine transformation

$$E_{\theta,\ell,t}(h_{\ell,t}) = h_{\ell,t} \odot \gamma_\ell + \beta_\ell, \quad \theta = (\gamma_\ell, \beta_\ell) \in \mathbb{R}^{1 \times 1 \times D_\ell} \times \mathbb{R}^{1 \times 1 \times D_\ell}, \quad (1)$$

where $\odot$ is the Hadamard product. Let $f_\ell(\cdot)$ denote the remainder network from layer $\ell$ to the task head, and let $\mathcal{L}(\cdot)$ be the task loss. We consider diagonal edits constrained to a feasible set $\mathcal{B}$ (e.g., $\|(\gamma_\ell - \mathbf{1}, \beta_\ell)\|_2 \leq \rho$ or box constraints on $\gamma_\ell$), which makes the optimization and approximation well-defined. For a codebook of $n$ editors $\Theta = \{\theta_i\}_{i=1}^n$, the token-wise selector returns a Top-$k$ index set $\mathcal{T} \subseteq \{1, \ldots, n\}, |\mathcal{T}| = k$, and nonnegative mixing weights $\{w_i\}_{i \in \mathcal{T}}$ with $\sum_{i \in \mathcal{T}} w_i = 1$. We write $\Theta_k = \text{conv}\{\theta_i : i \in \mathcal{T}\}$ for the corresponding convex hull. Unless stated otherwise, $\|\cdot\|$ denotes the Euclidean norm.

**Token-optimal diagonal edit.** For a fixed token representation $h_{\ell,t}$, we define the token-optimal diagonal parameters as

$$\theta^\star(h_{\ell,t}) \in \arg\min_{\theta \in \mathcal{B}} \mathcal{L}\big(f_\ell(E_{\theta,\ell,t}(h_{\ell,t}))\big). \quad (2)$$

This object serves as the ground-truth reference for our approximation analysis; it is the best diagonal edit (within $\mathcal{B}$) for the current token at layer $\ell$.

**Why token-aware edits are necessary.** Consider a first–order Taylor expansion of $\mathcal{L}\big(f_\ell(E_{\theta,\ell,t}(h_{\ell,t}))\big)$ around the identity edit $(\gamma_\ell, \beta_\ell) = (\mathbf{1}, \mathbf{0})$:

$$\mathcal{L}\big(f_\ell(E_{\theta,\ell,t}(h_{\ell,t}))\big) \approx \mathcal{L}\big(f_\ell(h_{\ell,t})\big) + \underbrace{g_\ell(h_{\ell,t})^\top\big((\gamma_\ell - \mathbf{1}) \odot h_{\ell,t} + \beta_\ell\big)}_{\text{first-order term}} + R_2(\theta; h_{\ell,t}), \quad (3)$$

where $g_\ell(h_{\ell,t}) = \nabla_{h_{\ell,t}} \mathcal{L}(f_\ell(h_{\ell,t}))$ and $R_2$ collects second-order terms (bounded under standard smoothness assumptions). Under a norm constraint on $(\gamma_\ell - \mathbf{1}, \beta_\ell)$, the first-order decrease aligns coordinate-wise with $-g_\ell(h_{\ell,t})$, which is token-dependent. Hence a single global edit is generally suboptimal; edits must be token-aware.

**Why a small set of prototypes suffices.** Empirically (Fig. 1), $\theta^\star(h_{\ell,t})$ across tokens clusters into a handful of smooth modes. This invites a codebook view: treat each editor parameter $\theta_i = (\gamma_i, \beta_i)$ as a codeword and the set $\Theta$ as a codebook. Classical vector quantization (e.g., Lloyd–Max, $k$-means) relates hard assignment (Top-1) error to within-cluster radius/variance; learning $\Theta$ reduces these radii, improving approximation of $\theta^\star(h_{\ell,t})$ by nearby codewords. We operationalize this with a token-wise selector.

**Why Top-$k$ convex mixing is principled.** Given the Top-$k$ set $\mathcal{T}$ and weights $\{w_i\}_{i \in \mathcal{T}}$, the mixed parameter is

$$\widehat{\theta} = \sum_{i \in \mathcal{T}} w_i \theta_i \in \Theta_k. \tag{4}$$

Let $\mathrm{dist}(\theta^\star, \Theta_k) = \min_{\vartheta \in \Theta_k} \|\theta^\star - \vartheta\|$ be the distance from the token-optimal parameter to the convex set $\Theta_k$. Then, by convexity,

$$\mathrm{dist}(\theta^\star, \Theta_k) \leq \min_{i \in \mathcal{T}} \|\theta^\star - \theta_i\|, \tag{5}$$

so allowing convex combinations (Top-$k$) is never worse than nearest-neighbor/Top-1 in parameter space.

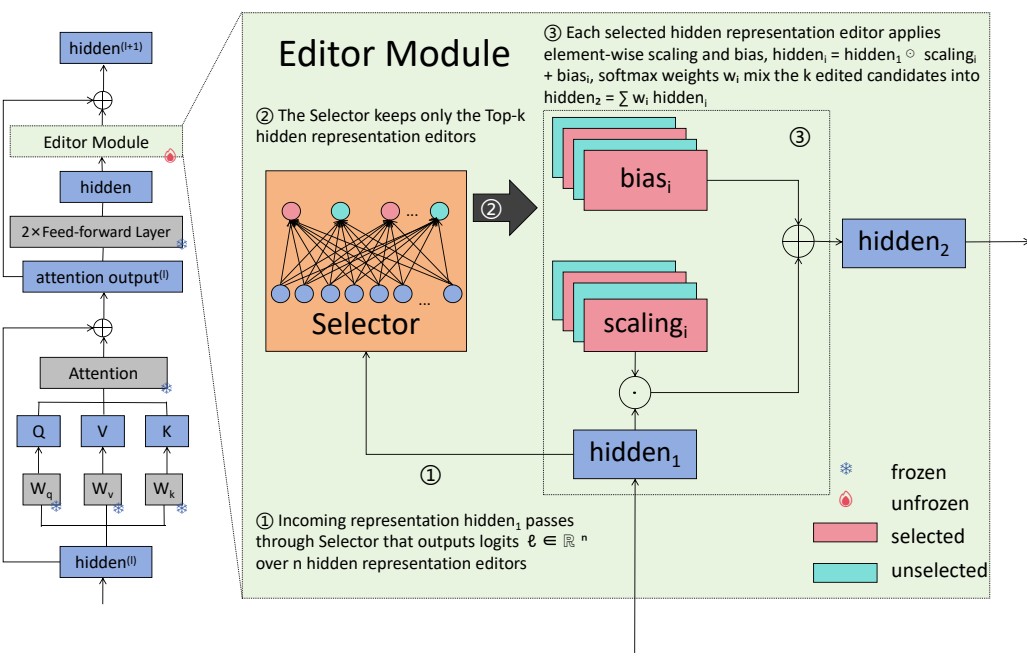

Figure 2: Schematic of the proposed TARE method.

**From parameter error to output error.** Fix $h_\ell$ and two parameter vectors $\theta, \theta'$. Since $E_{\theta,\ell,t}(h_{\ell,t}) = h_{\ell,t} \odot \gamma_\ell + \beta_\ell$ is affine in $\theta$, one has

$$\|E_{\theta,\ell,t}(h_{\ell,t}) - E_{\theta',\ell,t}(h_{\ell,t})\|_2 = \| h_{\ell,t} \odot (\gamma_\ell - \gamma'_\ell) + (\beta_\ell - \beta'_\ell) \|_2 \leq L(h_{\ell,t}) \|\theta - \theta'\|_2, \tag{6}$$

with $L(h_{\ell,t}) = \sqrt{\|h_{\ell,t}\|_\infty^2 + 1}$ (a token-dependent Lipschitz constant; proof in A.3). Combining equation 5 and equation 6 yields an end-to-end token-level bound:

$$\|E_{\widehat{\theta},\ell,t}(h_{\ell,t}) - E_{\theta^\star,\ell,t}(h_{\ell,t})\|_2 \leq L(h_{\ell,t}) \, \mathrm{dist}(\theta^\star, \Theta_k) \leq L(h_{\ell,t}) \min_{i \in \mathcal{T}} \|\theta^\star - \theta_i\|_2. \tag{7}$$

Thus, learning a small set of diagonal hidden representation editors (a codebook) and performing token-wise Top-$k$ convex mixing provides a principled approximation of the unknown token-optimal edit, with guarantees that are never worse than Top-1 and improve as the learned codewords shrink the cluster radii.

**Summary.** (1) The first-order analysis equation 3 motivates token-aware diagonal edits. (2) The clustering of $\theta^\star(h_{\ell,t})$ across tokens justifies a finite codebook of hidden representation editors. (3) Top-$k$ convex mixing is a principled realization, with the projection bound equation 5 and the Lipschitz link equation 7 connecting parameter-space approximation to output-space error. These results explain why TARE attains fine-grained adaptivity with near-zero inference overhead: hidden representation editors are diagonal (cheap) and selection is sparse (Top-$k$).

## 3.2 OVERALL DESIGN

The proposed TARE method augments hidden representation editor with a lightweight token-aware selector, as shown in Figure 2. At each token position, the selector activates a small subset of hidden representation editors, each providing its own per-feature scaling and bias; they are then linearly combined with normalized weights. This multi-path yet $k$-sparse design enables flexible and efficient token-wise adjustment, enhancing adaptability across heterogeneous tasks while keeping inference overhead negligible.

For every layer, TARE attach $n$ hidden representation editors, each with an independent parameter set for editing operations (element-wise scaling and bias by default, extensible to other simple transforms). During the forward pass, a Top-$k$ mechanism selects the $k$ most relevant hidden representation editors conditioned on the current activation, and the final representation is obtained by a weighted combination of their edits.

The proposed TARE method consists of three main steps: Token-Aware Selection, Top-$k$ Activation, and Hidden Representation Editing and Aggregation.

## 3.3 TOKEN-AWARE SELECTION

Let the hidden representation of a given token $t$ at layer $\ell$ be $h_{\ell,t} \in \mathbb{R}^{1 \times 1 \times D_\ell}$. TARE first applies a token-wise selector: a small feed-forward network that produces a real-valued score for each of the $n$ candidate diagonal editors. Formally,

$$h_{\ell,t}^{\text{new}} = \text{selector}(h_{\ell,t}) \quad \in \mathbb{R}^{1 \times 1 \times n}. \tag{8}$$

The selector uses one linear layer and is kept narrow so its parameter footprint remains negligible. Intuitively, it scores token–editor compatibility, playing a role analogous to a gating network while keeping the backbone frozen.

## 3.4 TOP-$k$ ACTIVATION

To avoid activating all $n$ hidden representation editors and increasing compute, The proposed TARE method keeps only the $k$ highest-scoring hidden representation editors per token ($k \ll n$, e.g., $k = 3$):

$$\big(\text{topk\_values, topk\_indices}\big) = \text{TopK}\big(h_{\ell,t}^{\text{new}}, k\big). \tag{9}$$

The selected logits are then normalized with a softmax (along the last dimension) to obtain a probabilistic selection mask:

$$w = \text{softmax}\big(\text{topk\_values}, -1\big), \tag{10}$$

so that $\sum_{i=1}^{k} w_{\ell,t,i} = 1$ for every token. This sparse selection keeps inference time virtually unchanged relative to the original model, because the cost of processing $k$ lightweight hidden representation editors is dominated by the backbone's already-computed attention and feed-forward layers.

The selector's Top-$k$ routing can collapse (most tokens routed to a few editors), which hurts both stability and capacity usage. We add a lightweight auxiliary term on the selector probabilities, which encourages balanced utilization across editors, stabilizes training, and yields consistent accuracy gains. A fuller discussion are given in A.4.

Table 1: Knowledge Reasoning results with LLaMA-3-8B.Results for LoRA, DoRA and LoReFT follow Wu et al. (2024b). MiLoRA numbers follow Wang et al. (2024a). LIFT numbers follow Liu et al. (2025). WeGeFT numbers follow Savadikar et al. (2025).

| PEFT | Source | Params.(%) | VRAM(MiB) | BoolQ | PIQA | SIQA | HellaS. | WinoG. | ARC-e | ARC-c | OBQA | Avg. |
|---|---|---|---|---|---|---|---|---|---|---|---|---|
| LoRA | ICLR 21 | 0.7002 | 21 828 | 70.8 | 85.2 | 79.9 | 91.7 | 84.3 | 84.2 | 71.2 | 79.0 | 80.8 |
| DoRA | ICML 24 | 0.7098 | 41 780 | 74.6 | 89.3 | 79.9 | 95.5 | 85.6 | 90.5 | 80.4 | 85.8 | 85.2 |
| MiLoRA | NAACL 25 | 0.7002 | 21 580 | 68.8 | 86.7 | 77.2 | 92.9 | 85.6 | 86.8 | 75.5 | 81.8 | 81.9 |
| LoReFT | NeurIPS 24 | 0.0260 | 21 050 | 75.1 | 90.2 | 82.0 | 96.3 | 87.4 | 92.4 | 81.6 | 87.5 | 86.6 |
| RED | ACL 24 | 0.0033 | 20 132 | 68.0 | 83.7 | 79.7 | 90.0 | 83.2 | 85.2 | 72.8 | 79.4 | 80.2 |
| LIFT | ICLR 25 | 5.0000 | 45 600 | 75.7 | 90.5 | 83.2 | **96.5** | 89.4 | **93.6** | 83.9 | **90.2** | **87.9** |
| WeGeFT | ICML 25 | 0.0130 | 20 364 | 75.7 | 89.9 | 82.5 | 96.4 | 88.7 | 92.5 | 82.3 | 86.3 | 86.8 |
| PiSSA | NeurIPS 24 | 0.7002 | 21 004 | 72.1 | 89.2 | 82.7 | 94.6 | 89.6 | 86.8 | **84.5** | 85.2 | 85.6 |
| Spectral Adapter | arXiv | 0.7002 | 21 746 | 72.1 | 88.3 | 83.1 | 94.6 | 89.3 | 85.4 | 82.2 | 85.2 | 85.0 |
| LoRA-GA | NeurIPS 24 | 0.7002 | 21 708 | 72.5 | 88.8 | 82.7 | 94.4 | 89.6 | 91.3 | 80.4 | 85.6 | 85.7 |
| LoRA-One | arXiv | 0.7002 | 21 206 | 72.0 | 88.9 | 82.9 | 94.4 | **89.8** | 85.1 | 82.6 | 87.6 | 85.4 |
| **TARE (ours)** | This paper | 0.0392 | 21 724 | 75.2 | 90.2 | 82.5 | 94.1 | 88.6 | 91.3 | 82.3 | 88.4 | 86.7 |
| **TARE (all)** | This paper | 0.4097 | 24 044 | **76.3** | **91.6** | **83.6** | 95.5 | **89.8** | 92.7 | 83.9 | 89.2 | 87.8 |

## 3.5 HIDDEN REPRESENTATION EDITING AND AGGREGATION

Each hidden representation editor $i$ maintains its own pair of element-wise scaling and bias vectors $\gamma_{\ell,i}, b_{\ell,i} \in \mathbb{R}^{1 \times 1 \times D_\ell}$, trained from scratch while the backbone remains frozen. For each selected hidden representation editor, the proposed TARE method compute a candidate edit

$$h_{\ell,t,i} = h_{\ell,t} \odot \gamma_{\ell,i} + b_{\ell,i}, \tag{11}$$

where $\odot$ denotes the Hadamard (element-wise) product. Because these operations are diagonal in feature space, they introduce no additional matrix multiplications and can be fused into a single CUDA kernel in practical implementations. Finally, the $k$ token-specific hidden representation editors are linearly combined according to their selection weights to yield the updated representation

$$h_{\ell,t}^{\text{update}} = \sum_{i=1}^{k} h_{\ell,t,i}\, w_{\ell,t,i}. \tag{12}$$

This convex combination acts as a soft winner-take-all mechanism: hidden representation editors that the selector deems most relevant contribute the most, while others are softly suppressed.

In summary, the proposed TARE method adds a lightweight, token-aware, $k$-sparse hidden representation editor that lifts the representational ceiling of simple scaling/bias edits while keeping the backbone frozen. By conditionally selecting and mixing a few per-feature edits per token, it attains high expressiveness and contextual adaptivity with near-zero inference overhead.

## 4 EXPERIMENT

Our work conduct a comprehensive study on decoder model LLaMA-3-8B and encoder model RoBERTa-base/large.The evaluation spans nine task families—knowledge reasoning, mathematical reasoning, GLUE, conditional text generation, code synthesis, knowledge completion, closed-book QA, symbolic reasoning and instruction following—against strong PEFT baselines (LoRA, DoRA, MiLoRA, LoReFT, RED; on GLUE our work also include Adapter-FFN, IA$^3$, and BitFit). Ablation Study isolate scaling vs. bias, and Sensitivity analysis study the number of hidden representation editors $n$ and the number of selected hidden representation editors $k$, quantifying the expressiveness–efficiency trade-off.In addition, a visualize analysis examines load-balancing behavior at the layer level, showing how the auxiliary loss equalizes editor utilization and correlates with consistent accuracy gains.For completeness, an expanded discussion of dataset,baseline and implementation detail is deferred to A.5, A.6 and A.7.

## 4.1 OVERALL PERFORMANCE

The proposed TARE method delivers state-of-the-art or competitive results across diverse tasks—including conditional text generation, code synthesis, knowledge reasoning, mathematical reasoning, GLUE, knowledge completion, closed-book QA and symbolic reasoning—while training

Table 2: Mathematical Reasoning results with LLaMA-3-8B and Qwen-2.5-7B-Instruct.

| PEFT | Source | Model | Params.(%) | VRAM(MiB) | MultiArith | GSM8k | SVAMP | MAWPS | AddSub | AQuA | SingleEq | Avg. |
|------|--------|-------|-----------|-----------|-----------|-------|-------|-------|--------|------|----------|------|
| LoRA | ICLR 21 | | 0.2345 | 21 070 | 92.0 | **61.4** | 69.8 | 84.2 | 85.4 | **44.7** | 90.3 | 75.4 |
| LoRA (all) | ICLR 21 | | 1.0338 | 24 622 | 95.5 | 57.5 | 69.4 | 86.5 | **91.2** | 41.3 | **93.3** | 76.4 |
| DoRA | ICML 24 | | 0.2361 | 29 284 | 91.7 | 59.0 | 72.3 | 82.1 | 86.1 | 39.9 | 89.5 | 74.4 |
| MiLoRA | NAACL 25 | LLaMA-3-8B | 0.2345 | 21 520 | 91.7 | 59.0 | 70.5 | **88.3** | 86.1 | 43.4 | 90.5 | 75.6 |
| LoReFT | NeurIPS 24 | | 0.0260 | 21 940 | 89.2 | 56.2 | 68.7 | 80.3 | 90.1 | 33.1 | 90.0 | 72.5 |
| RED | ACL 24 | | 0.0033 | 19 852 | 91.0 | 54.2 | 66.8 | 81.1 | 87.3 | 34.1 | 90.9 | 72.2 |
| **TARE (ours)** | This paper | | 0.0392 | 20 900 | **95.8** | 57.3 | **72.9** | 86.1 | **90.9** | 41.4 | 92.1 | **76.7** |
| LoRA | ICLR 21 | | 0.2643 | 21 244 | 94.7 | 72.8 | **81.1** | 89.4 | 88.4 | 66.5 | 91.7 | 83.5 |
| DoRA | ICML 24 | | 0.2657 | 29 604 | 93.2 | 72.1 | 79.8 | 88.2 | 89.7 | 63.7 | **92.7** | 82.8 |
| MiLoRA | NAACL 25 | Qwen-2.5-7B-Instruct | 0.2643 | 21 518 | 93.3 | 72.2 | 80.8 | 88.3 | 89.6 | 69.6 | **92.7** | 83.8 |
| LoReFT | NeurIPS 24 | | 0.0218 | 21 832 | 92.1 | 71.7 | 78.5 | 86.2 | 90.0 | 67.3 | 90.5 | 82.3 |
| RED | ACL 24 | | 0.0026 | 20 100 | 91.3 | 71.3 | 77.4 | 84.1 | **90.4** | 70.9 | 88.2 | 81.9 |
| **TARE (ours)** | This paper | | 0.0316 | 20 624 | **96.0** | **75.1** | 80.3 | **92.4** | **90.4** | 63.6 | 91.3 | **84.2** |

only 0.0392% of parameters and maintaining low VRAM with near-zero inference overhead (e.g., E2E best on all metrics; HumanEval/MBPP highest Pass@1 Rate; Commonsense avg. 86.7%; Math-10K avg. 76.7%; GLUE 88.3%), outperforming or matching LoRA/DoRA/MiLoRA/LoReFT/RED.

#### 4.1.1 KNOWLEDGE REASONING

TARE attains an average accuracy of 86.7 on the eight commonsense-reasoning benchmarks in Table 1, placing it in the top tier of PEFT methods. Although the heavy LIFT model reaches the highest average of 87.9, it requires 5.0% trainable parameters and 45,600 MiB VRAM, whereas TARE is only 0.0392% ($\sim$ 1/128 as many parameters) and 21,724 MiB VRAM, yet trails by just 1.2 points. Compared with other strong PEFT baselines, TARE improves the average accuracy over LoRA, MiLoRA, RED, DoRA, PiSSA, Spectral Adapter, LoRA-GA, and LoRA-One by +5.9, +4.8, +6.5, +1.5, +1.1, +1.7, +1.0, and +1.3 points, respectively, and slightly edges out LoReFT by +0.1 and is essentially on par with the recent WeGeFT method (86.8). Across individual datasets, TARE remains consistently close to the best-performing methods—for example, 90.2 on PIQA, 94.1 on HellaSwag, 88.6 on WinoGrande, 82.3 on ARC-c, and 88.4 on OBQA—while using two orders of magnitude fewer trainable parameters than most LoRA-style variants, highlighting a favorable accuracy–efficiency trade-off. When we allow TARE to adapt all projection matrices $q/k/v/o$ and $up/gate/down$ (TARE (all) in Table 1), the average accuracy further improves to 87.8, essentially matching the heavy LIFT model (87.9) while remaining much more efficient. Concretely, TARE (all) uses only 0.4097% trainable parameters (about $12\times$ fewer than LIFT's 5.0%) and 24,044 MiB VRAM (vs. 45,600 MiB for LIFT).

#### 4.1.2 MATHEMATICAL REASONING

TARE consistently attains the highest average accuracy on the seven math-reasoning benchmarks in Table 2 for both backbones. On LLaMA-3-8B, it reaches an average accuracy of 76.7 with only 0.0392% trainable parameters and 20,900 MiB peak VRAM, achieving the best results on MultiArith (95.8), SVAMP (72.9), AddSub (90.9), and SingleEq (92.1), and remaining competitive on GSM8k (57.3), MAWPS (86.1), and AQuA (41.4). Even compared with the much heavier LoRA (all), which applies LoRA to all seven projection matrices with 1.0338% trainable parameters and 24,622 MiB VRAM, TARE attains a higher average accuracy (76.7 vs. 76.4), and on average improves over LoRA / LoRA (all) / MiLoRA / DoRA / RED / LoReFT by (+1.3 / +0.3 / +1.1 / +2.3 / +4.5 / +4.2) points while being far more parameter-efficient than all low-rank baselines. On Qwen-2.5-7B-Instruct, TARE further achieves the best average accuracy of 84.2 with only 0.0316% trainable parameters and 20,624 MiB peak VRAM, obtaining the highest or tied-highest scores on MultiArith (96.0), GSM8k (75.1), MAWPS (92.4), and AddSub (90.4), and improving over LoRA / MiLoRA / DoRA / RED / LoReFT by (+0.7 / +0.4 / +1.4 / +2.3 / +1.9) points on average. For a fair comparison, all PEFT methods except LoRA (all) are applied only to the projection layer of the MLP blocks.

#### 4.1.3 GLUE (A.8)

#### 4.1.4 CONDITIONAL TEXT GENERATION

TARE achieves the best E2E conditional generation with LLaMA-3-8B, reaching BLEU 0.6333, NIST 8.3105, METEOR 0.4456, ROUGE–L 0.6758, and CIDEr 2.2027 in Table 3. It surpasses

Table 3: Conditional Text Generation results with LLaMA-3-8B.

| PEFT | Source | Params.(%) | VRAM(MiB) | BLEU↑ | NIST↑ | METEOR↑ | ROUGE–L↑ | CIDEr↑ |
|---|---|---|---|---|---|---|---|---|
| LoRA | ICLR 21 | 0.2345 | 39 166 | 0.6255 | 8.2791 | 0.4404 | 0.6661 | 2.1524 |
| DoRA | ICML 24 | 0.2361 | 45 326 | 0.6201 | 8.1455 | 0.4367 | 0.6617 | 2.1578 |
| MiLoRA | NAACL 25 | 0.2345 | 39 590 | 0.6244 | 8.2652 | 0.4283 | 0.6606 | 2.1845 |
| LoReFT | NeurIPS 24 | 0.0260 | 32 502 | 0.5719 | 7.5671 | 0.4304 | 0.6431 | 1.6881 |
| RED | ACL 24 | 0.0033 | 29 492 | 0.5994 | 7.9229 | 0.4401 | 0.6692 | 2.1958 |
| **TARE (ours)** | This paper | 0.0392 | 34 626 | **0.6333** | **8.3105** | **0.4456** | **0.6758** | **2.2027** |

Table 4: Instruction Following results with LLaMA-2-7B. Results for LoRA, PiSSA, rsLoRA and LoRA+ follow Wang et al. (2024c).

| PEFT | Source | Params.(%) | First Turn Score |
|---|---|---|---|
| LoRA | ICLR 21 | 0.2970 | $5.61 \pm 0.10$ |
| PiSSA | NeurIPS 24 | 0.2970 | $5.30 \pm 0.02$ |
| rsLoRA | arXiv | 0.2970 | $5.25 \pm 0.03$ |
| LoRA+ | ICML 24 | 0.2970 | $5.71 \pm 0.08$ |
| **TARE (ours)** | This paper | 0.0467 | $\mathbf{5.73 \pm 0.05}$ |

the strongest baselines on each metric, for example by about +0.008 BLEU over LoRA (0.6255), +0.031 NIST over LoRA (8.2791), +0.005 METEOR over LoRA (0.4404), +0.0066 ROUGE–L over RED (0.6692), and +0.0069 CIDEr over RED (2.1958). The method trains only 0.0392% of parameters and uses 34,626 MiB peak VRAM, thus delivering higher text quality while remaining highly parameter efficient and lighter than LoRA and DoRA in memory usage. All PEFT methods are applied to the projection layer of the MLP blocks in the backbone language model.

### 4.1.5 CODE SYNTHESIS (A.9)

### 4.1.6 KNOWLEDGE COMPLETION (A.10)

### 4.1.7 CLOSED-BOOK QA AND SYMBOLIC REASONING (A.11)

### 4.1.8 INSTRUCTION FOLLOWING

From Table 4, under the setting where LLaMA-2-7B is instruction-tuned on WizardLM and evaluated on MT-Bench with GPT-4 scoring, TARE achieves the best First Turn Score with extremely low parameter overhead: LoRA, PiSSA, rsLoRA, and LoRA+ each require (0.297%) trainable parameters, whereas TARE uses only (0.0467%) yet still attains the highest score of $(5.73 \pm 0.05)$, surpassing the strongest baseline LoRA+ $(5.71 \pm 0.08)$ and clearly outperforming PiSSA $(5.30 \pm 0.02)$ and rsLoRA $(5.25 \pm 0.03)$. This shows that, under identical training data and evaluation protocols, TARE learns more robust instruction-alignment behaviour within a much smaller update space, making first-turn responses to open and complex instructions more aligned with human preferences and less prone to failure, and providing a stronger basis for context understanding and task decomposition in subsequent multi-turn interactions. Meanwhile, its tiny parameter ratio reduces deployment costs and the risk of catastrophic forgetting, making incremental enhancement or domain extension of deployed dialogue agents safer and more efficient, and highlighting the practicality and robustness of TARE in long-conversation and online-service settings.

### 4.2 ABLATION STUDY

Component ablation. TARE attains the best overall result, reaching an average accuracy of 76.7 with only 0.0392% trainable parameters and about 20,900 MiB peak VRAM (Table 13). It clearly outperforms both ablated variants—w/o scaling (50.5) and w/o bias (56.4). On representative datasets, the full scaling,+,bias edit delivers large gains: MultiArith +35–52,pp, GSM8k +24–29,pp, SVAMP ≈,+17,pp, MAWPS +31–33,pp, AddSub +10–20,pp, AQuA +9–10,pp, and SingleEq +15–23,pp over the ablations. These improvements match the design intent: per-dimension scaling calibrates

feature magnitudes, per-dimension bias corrects offsets, and their joint, token-wise adjustment better aligns hidden representations with task signals.

Table 5: Position ablation of TARE on LLaMA-3-8B.

| PEFT | Params.(%) | VRAM(MiB) | MultiArith | GSM8k | SVAMP | MAWPS | AddSub | AQuA | SingleEq | Avg. |
|---|---|---|---|---|---|---|---|---|---|---|
| **TARE (q)** | 0.0392 | 20 430 | 86.0 | 50.4 | 63.3 | 74.8 | 77.5 | 36.7 | 86.8 | 67.9 |
| **TARE (k)** | 0.0098 | 18 492 | 78.3 | 44.7 | 60.4 | 73.9 | 78.2 | **43.1** | 85.6 | 66.3 |
| **TARE (v)** | 0.0098 | 18 486 | 91.0 | 56.0 | 68.1 | 79.4 | 86.1 | 38.4 | 90.4 | 72.8 |
| **TARE (o)** | 0.0392 | 22 438 | 92.0 | 57.9 | 72.2 | 85.3 | 88.6 | 39.4 | 92.1 | 75.4 |
| **TARE (up)** | 0.1369 | 29 556 | 91.7 | **62.2** | 69.6 | **88.2** | 87.3 | 38.4 | **92.3** | 75.7 |
| **TARE (gate)** | 0.1369 | 29 536 | 87.0 | 54.7 | 67.5 | 82.8 | 85.1 | 32.6 | 91.7 | 71.6 |
| **TARE (down)** | 0.0392 | 20 900 | **95.8** | 57.3 | **72.9** | 86.1 | **90.9** | 41.4 | 92.1 | **76.7** |

**Load-balancing ablation.** Adding the load-balancing auxiliary loss(A.4) yields a higher average accuracy of 76.7 vs. 75.8 without it, at the same 0.0392% trainable ratio and nearly unchanged VRAM (Table 14; a detailed description of this loss is provided in the Appendix). The loss prevents routing collapse and spreads token traffic across editors: in the 16th block, selection counts move from highly skewed—one editor rarely chosen and others around $7.3\times10^5$–$1.16\times10^6$—to near-uniform use of all eight editors ($\sim 7.6\times10^5$–$8.2\times10^5$ each), as shown in Fig. 3. This fuller capacity utilization translates into consistent metric gains, e.g., MultiArith $+2.0$, AddSub $+2.0$, and AQuA $+1.6$ (Table 14), because more balanced routing exposes diverse tokens to specialized diagonal edits without adding parameters or inference cost.

**Position ablation.** Table 5 reports the performance of TARE when inserted at seven locations in LLaMA-3-8B—self-attention projections q/k/v/o and FFN linear layers up/gate/down—showing clear differences in effectiveness and cost. Applying TARE to q and k yields very low trainable parameter ratios (0.0392%/0.0098%) and VRAM (20.4/18.5 GiB), but poor average accuracies (67.9/66.3). Moving TARE to v improves the average to 72.8, indicating that editing value vectors is more effective than perturbing queries/keys. Inserting it on o further raises the average to 75.4, with strong scores on several math datasets, at the cost of higher VRAM (22.4 GiB). For the FFN, attaching TARE to up attains an average of 75.7 and excels on GSM8k and MAWPS, but requires 0.1369% trainable parameters and about 29.6 GiB VRAM, while gate is weaker overall (average 71.6). In contrast, placing TARE on the FFN down layer (our default) offers the best accuracy–efficiency trade-off: with only 0.0392% trainable parameters and 20.9 GiB VRAM, it achieves the highest average accuracy of 76.7 and near-best or best scores on multiple datasets. This shows that editing the down-projection layer best exploits FFN nonlinearity while preserving excellent parameter and memory efficiency, making it the most effective insertion point for TARE.

### 4.3 SENSITIVITY ANALYSIS

**Hyperparameter sensitivity.** TARE achieves its best average accuracy of 76.7% with $k=3$ selected hidden representation editors (Fig. 4, Table 15). Increasing $k$ from 1 to 3 sharply improves accuracy (71.2% $\rightarrow$ 76.7%), while larger $k$ yields diminishing returns and task-specific peaks (e.g., GSM8k at $k=7$, AQuA at $k=8$, SingleEq at $k=6$), suggesting that too many edits over-average token signals whereas a small set captures the dominant modes. Memory grows only modestly as $k$ increases ($\approx 20.2$ GiB at $k=1$ to $\approx 22.8$ GiB at $k=8$) with the trainable-parameter ratio fixed at 0.0392%, so $k=3$ provides a strong accuracy–efficiency trade-off and is our default choice. We further vary the total number of editors $n$ with $k=3$ fixed (Table 16). TARE remains stable for $n \in 6, 8, 10$: the average accuracy varies within 0.6 points and peaks at 76.7% when $n=8$, slightly above $n=6$ (76.5%) and $n=10$ (76.1%). Different tasks favor slightly different $n$, but a moderately overcomplete pool ($n=8$) already covers diverse reasoning modes: further increasing $n$ offers only marginal gains while raising the trainable-parameter ratio (0.0294%,$\rightarrow$,0.0489%) with nearly unchanged VRAM, so we adopt $n=8$ as the default to balance robustness and parameter efficiency.

**Sample sensitivity.** Table 6 reports TARE on LLaMA-3-8B when fine-tuned on different numbers of Math-10k training examples (500, 1,000, 2,000, 5,000, 9919 (all)) under the same backbone and hyperparameter setting (always 0.0392% trainable parameters and about 20 GiB VRAM). We observe stable and robust performance across supervision scales: even with 500 examples, TARE

Table 7: Comparison between LoRA and TARE in terms of parameter ratio, training time, inference time statistics on LLaMA-3-8B.

| PEFT | Source | Params. (%) | Training time (s/epochs) | Mean inference time (s) |
|------|--------|-------------|--------------------------|--------------------------|
| LoRA | ICLR 21 | 0.2345 | 1015.56 | **2.68 ± 0.43** |
| **TARE (ours)** | This paper | 0.0392 | **837.35** | 3.20 ± 0.67 |

Table 8: Token-wise selection and editing time statistics of TARE on LLaMA-3-8B.

| PEFT | Mean selection time(s) | Mean editing time(s) |
|------|------------------------|----------------------|
| **TARE (ours)** | $6.94 \times 10^{-5} \pm 1.46 \times 10^{-8}$ | $9.78 \times 10^{-5} \pm 1.31 \times 10^{-10}$ |

reaches an average accuracy of 69.8 over seven math benchmarks, and as the sample size increases to 2,000 and 5,000, the average accuracy rises smoothly to 74.3 and 76.2 (e.g., GSM8k 60.8 at 5,000), without training instability or large fluctuations. Using the full Math-10k further boosts the average to 76.7 (with strong results on MultiArith, SVAMP, AddSub, etc.), showing that TARE is already effective in the low-data regime and continues to improve as more data become available, indicating a stable, well-generalised token-selection and editing strategy rather than reliance on massive labelled data.

Table 6: Sample Sensitivity Analysis of TARE on LLaMA-3-8B.

| PEFT | Params.(%) | VRAM(MiB) | MultiArith | GSM8k | SVAMP | MAWPS | AddSub | AQuA | SingleEq | Avg. |
|------|-----------|-----------|------------|-------|-------|-------|--------|------|----------|------|
| **TARE (sample=500)** | 0.0392 | 20 398 | 86.3 | 52.5 | 64.1 | 81.1 | 81.0 | 36.7 | 87.0 | 69.8 |
| **TARE (sample=1000)** | 0.0392 | 20 406 | 85.3 | 51.3 | 68.6 | 81.1 | 85.3 | 32.8 | 89.4 | 70.5 |
| **TARE (sample=2000)** | 0.0392 | 20 606 | 89.0 | 56.6 | 67.5 | 84.9 | 85.1 | **46.5** | 90.6 | 74.3 |
| **TARE (sample=5000)** | 0.0392 | 20 274 | 91.8 | **60.8** | 68.1 | **87.4** | 88.9 | 41.9 | **94.3** | 76.2 |
| **TARE (sample=9919)** | 0.0392 | 20 900 | **95.8** | 57.3 | **72.9** | 86.1 | **90.9** | 41.4 | 92.1 | **76.7** |

## 4.4 EFFICIENCY ANALYSIS

From Table 7, under the same backbone (LLaMA-3-8B) and task setup, TARE markedly improves training and inference efficiency while incurring almost no extra inference cost. In terms of parameter scale, TARE uses only 0.0392% trainable parameters, reducing updated weights by about five sixths compared with LoRA's 0.2345%, which directly lightens backpropagation and optimization. On Math-10K, this leads to a shorter per-epoch training time (1015.56 s for LoRA vs. 837.35 s for TARE, a ∼17% reduction), yielding substantial compute savings for long-horizon fine-tuning. At inference, although LoRA can merge its low-rank weights and is theoretically zero-overhead, its empirical mean latency on 600 MultiArith examples is 2.68 s (var. 0.43), while TARE—without weight merging and preserving online editing—achieves a very similar 3.20 s (var. 0.67), only about 19% higher and well below one extra second per sample. Finer-grained measurements (Table 8) show that TARE's token-level selection and editing are almost free: mean selection time $6.94 \times 10^{-5}$ s (var. $1.46 \times 10^{-8}$) and mean editing time $9.78 \times 10^{-5}$ s (var. $1.31 \times 10^{-10}$), i.e., sub-millisecond overhead. Overall, TARE offers a smaller parameter footprint, faster training, and near-zero additional inference cost, demonstrating high efficiency and practicality for real-world deployment.

## 5 CONCLUSION

Our work presented **Token-Aware Representation Editing** (*TARE*), a lightweight PEFT approach that replaces one-size-fits-all edits with per-token, per-dimension adjustments. Extensive experiments validate TARE's benefits on both decoder and encoder families, while tuning only 0.0392% of parameters and using about 20GiB of GPU memory, matching or surpassing LoRA, DoRA, MiLoRA, LoReFT, and RED across many settings.

## ETHICS STATEMENT

We affirm that all authors have read and will adhere to the ICLR Code of Ethics (`https://iclr.cc/public/CodeOfEthics`). The Code applies to every stage of our participation, including submission, discussion, and (if applicable) reviewing.

**Human subjects and privacy.** Our work does not involve user studies, human participants, or the collection of personally identifiable information. All experiments use publicly available datasets under their respective licenses. We do not attempt to deanonymize data or link records across datasets. When datasets include potentially sensitive content (e.g., natural language containing demographic references), we use them solely for research benchmarking and follow their intended-use guidelines.

**Data governance and licenses.** We respect dataset licenses and attribution requirements. Any data filtering or preprocessing is documented in the paper or appendix to support transparency and reproducibility. We do not redistribute third-party datasets; readers should obtain them from the original sources under the original terms.

**Safety, misuse, and downstream impacts.** The proposed TARE method is a generic fine-tuning technique that can improve model adaptability. Like other PEFT methods, it could be applied to harmful tasks if misused. We do not target such applications and discourage any use that violates the Code of Ethics or applicable laws. If we release code and scripts, we will include a model card and usage guidelines clarifying intended use, out-of-scope use cases, and safety considerations. We also encourage practitioners to implement content filtering and abuse monitoring when deploying fine-tuned models.

**Bias, fairness, and representational harms.** Large language models can reflect and amplify biases present in training data. While our work focuses on parameter efficiency rather than content shaping, improved adaptation can inadvertently strengthen biased behaviors inherited from data. We therefore report results across diverse task families and discuss limitations. We recommend additional fairness evaluations and domain-specific audits before deployment, especially in high-stakes settings.

**Security and legal compliance.** We do not circumvent access controls or use prohibited sources. All experiments comply with the terms of service of data and model providers and with applicable intellectual-property and data-protection laws.

**Reproducibility and transparency.** We describe datasets, model backbones, hyperparameters, and compute settings in the paper or appendix. Upon acceptance, we plan to release code, configuration files, and instructions to reproduce the main results, subject to license constraints of any third-party assets.

**Conflicts of interest and sponsorship.** The authors disclose no conflicts of interest beyond those stated in the metadata of the submission. No external sponsorship influenced the results or their presentation beyond acknowledged funding (if any) in the paper.

**Environmental considerations.** To reduce computational footprint, we use parameter-efficient fine-tuning and `bfloat16` precision. We encourage practitioners to reuse our released checkpoints and scripts, and to select smaller backbones when appropriate.

This ethics statement is provided to proactively address potential concerns regarding data practices, fairness, safety, reproducibility, and compliance. We welcome reviewer feedback on any additional considerations relevant to the ICLR Code of Ethics.

## REPRODUCIBILITY STATEMENT

We take several steps to facilitate independent verification of our results. The core algorithmic design of TARE are specified in §3 (with ablations and sensitivity analyses in §4.2 and §4.3). Datasets,

splits, and evaluation metrics are summarized in Table 9 and further detailed in A.5. Implementation particulars (model backbones, precision, optimizer, batch size, and hardware) are provided in A.6 and A.7. Theoretical clarifications and auxiliary loss formulations appear in A.4. Together, these materials are intended to enable end-to-end replication of our pipelines and numerical results.

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

# A APPENDIX

## A.1 USE OF LLMs

We used large language models strictly for editorial assistance, including spell checking, grammar polishing, and minor wording suggestions for the paper text. No model outputs were used to create, modify, or label datasets, implement algorithms, tune hyperparameters, or select results. All technical content (methods, proofs, experiments, and numbers) was written and verified by the authors, and every LLM-suggested edit was reviewed manually for accuracy and clarity.

## A.2 RELATED WORK

**Parameter-Efficient Fine-Tuning and Representation Editing** PEFT aims to adapt large Transformers by training only a tiny fraction of parameters while freezing the backbone. Low-rank adapters such as LoRA Hu et al. (2021) inject rank-$r$ updates into weight matrices; in standard single-adapter deployments these increments are merged into the base weights, so there is effectively no additional inference overhead. They still require nontrivial choices of rank and scaling, which can complicate tuning across layers and tasks. When merging is not desirable (e.g., multi-adapter hot-swap, online mixture/selection, or coexistence with certain quantization pipelines), one may resort to on-the-fly composition that re-introduces extra operators, but this is an engineering choice rather than an inherent property of LoRA-style methods. DoRA Liu et al. (2024) decouples direction and magnitude to stabilize optimization while remaining low-rank; MiLoRA Wang et al. (2024a) modifies singular subspaces to reduce redundancy in LoRA updates. A complementary line edits hidden representations directly: RED Wu et al. (2024a) learns a single global diagonal scaling/bias with near-zero inference cost but limited contextual adaptivity; LoReFT Wu et al. (2024b) performs low-rank projections in representation space but applies the same projection to every token and inherits rank selection. Our work follows representation editing but replaces one-size-fits-all edits with token-aware diagonal modulation, retaining the efficiency of feature-wise operations while addressing the lack of per-token expressiveness observed in global edits and uniform low-rank mappings.

**Token-Aware Conditional Modulation and Dynamic Editing** For encoder models, widely used PEFT baselines include LoRA and RED as above, together with IA$^3$ Liu et al. (2022) and BitFit Ben Zaken et al. (2021). IA$^3$ gates attention/FFN channels via learned per-feature multipliers, and BitFit updates only biases; both are extremely lightweight but share a uniform modulation across tokens, limiting fine-grained adaptivity. RED is inference-friendly but globally shared. In contrast, the proposed TARE method performs token-aware, diagonal representation editing: for each token it mixes a few learned diagonal templates to yield per-token, per-dimension adjustments while preserving near-zero inference overhead. This design directly targets the expressiveness–efficiency tension highlighted by these baselines. You may include other additional sections here.

**Relation to Mixture-of-Experts (MoE)** Similarities. TARE borrows two well-established ideas from the MoE literature Fedus et al. (2022): (i) token-wise sparse routing, where each token is routed to a small subset (Top-$k$) of candidates; and (ii) an auxiliary load-balancing loss that encourages the average routing distribution to be close to uniform, preventing collapse of routing to only a few choices. In our implementation the selector produces token-level scores over $n$ candidates and activates $k$ of them, and we use a KL-to-uniform load-balancing term (weight $\lambda{=}0.02$) to distribute traffic across candidates. Key differences. Despite these conceptual overlaps, TARE is not an MoE replacement of FFN layers. In classic MoE, each "expert" is a full (or sizable) feed-forward subnetwork that replaces the FFN block for routed tokens, incurring additional matmuls, parameters, capacity management, and dispatch overhead at inference. By contrast, TARE's "experts" are lightweight diagonal hidden representation editors—per-dimension scale and bias applied after the FFN within a PEFT regime. The backbone remains frozen; no FFN is duplicated or replaced. Computation stays strictly diagonal and sparse, yielding near-zero inference overhead and a parameter

footprint ($\ll 1\%$) in line with PEFT goals. Practically, TARE performs a convex mixture of a few diagonal edits for each token rather than switching among large FFN experts, so there is no capacity factor tuning or expert-capacity drop, and routing latency is negligible. Positioning and intent. We intentionally reuse MoE's load-balancing principle to stabilize token-wise routing and improve utilization, while introducing a new application of these principles to efficient representation editing. This framing positions TARE as a creative specialization of MoE-style routing for PEFT: it preserves the benefits of token-level adaptivity, but delivers them through tiny diagonal hidden representation editors that are computationally frugal and architecturally compatible with frozen backbones and low-overhead fine-tuning.

## A.3 LIPSCHITZ CONTINUITY OF THE EDITOR'S PARAMETERS

Fix a layer $\ell$ and a token $t$'s hidden representation $h_{\ell,t} \in \mathbb{R}^{1 \times 1 \times D_\ell}$. For diagonal hidden representation editors $E_\theta(h_{\ell,t}) = h_{\ell,t} \odot \gamma_\ell + \beta_\ell$ with $\theta = (\gamma_\ell, \beta_\ell) \in \mathbb{R}^{1 \times 1 \times D_\ell} \times \mathbb{R}^{1 \times 1 \times D_\ell}$, we have for any $\theta, \theta'$:

$$\left\| E_{\theta,\ell,t}(h_{\ell,t}) - E_{\theta',\ell,t}(h_{\ell,t}) \right\|_2 \ \leq \ L(h_{\ell,t}) \left\| \theta - \theta' \right\|_2, \qquad L(h_{\ell,t}) := \sqrt{\|h_{\ell,t}\|_\infty^2 + 1}. \tag{13}$$

Let $\Delta\gamma_\ell := \gamma_\ell - \gamma_\ell'$ and $\Delta\beta_\ell := \beta_\ell - \beta_\ell'$, and write $\Delta\theta := (\Delta\gamma_\ell, \Delta\beta_\ell)$. By definition,

$$E_{\theta,\ell,t}(h_{\ell,t}) - E_{\theta',\ell,t}(h_{\ell,t}) = h_{\ell,t} \odot \Delta\gamma_\ell + \Delta\beta_\ell. \tag{14}$$

Using the triangle inequality and Hölder/Cauchy–Schwarz,

$$\left\| h_{\ell,t} \odot \Delta\gamma_\ell + \Delta\beta_\ell \right\|_2 \ \leq \ \|h_{\ell,t} \odot \Delta\gamma_\ell\|_2 + \|\Delta\beta_\ell\|_2 \ \leq \ \|h_{\ell,t}\|_\infty \|\Delta\gamma_\ell\|_2 + \|\Delta\beta_\ell\|_2. \tag{15}$$

Define $u := (\|h_{\ell,t}\|_\infty, 1) \in \mathbb{R}^2$ and $v := (\|\Delta\gamma_\ell\|_2, \|\Delta\beta_\ell\|_2) \in \mathbb{R}^2$. Then the previous line is $u^\top v$ and, by Cauchy–Schwarz,

$$u^\top v \ \leq \ \|u\|_2 \|v\|_2 = \sqrt{\|h_{\ell,t}\|_\infty^2 + 1} \ \sqrt{\|\Delta\gamma_\ell\|_2^2 + \|\Delta\beta_\ell\|_2^2} = L(h_{\ell,t}) \|\Delta\theta\|_2. \tag{16}$$

This proves the claim.

## A.4 LOAD-BALANCING AUXILIARY LOSS

Let $N = B \times L$ be the number of tokens in a batch, and let $p_t \in \Delta^{n-1}$ denote the token-wise selection distribution over the $n$ hidden representation editors (e.g., the softmax over the last dimension of $h_1^{\text{new}}$; it may be computed on the Top-$k$ subset or on all $n$ hidden representation editors). Our work define the average selection distribution across tokens

$$\bar{p} \ = \ \frac{1}{N} \sum_{t=1}^{N} p_t \ \in \ \Delta^{n-1}, \tag{17}$$

and the uniform distribution $U = (\frac{1}{n}, \dots, \frac{1}{n})$. The load-balancing regularizer encourages aggregate editor usage to be uniform by minimizing the KL divergence

$$\begin{aligned}
\mathcal{L}_{\text{LB}} \ &= \ \lambda \, \text{KL}\big(\bar{p} \, \| \, U\big) \\
&= \ \lambda \sum_{i=1}^{n} \bar{p}_i \log \frac{\bar{p}_i}{1/n} \\
&= \ \lambda\Big( \sum_{i=1}^{n} \bar{p}_i \log \bar{p}_i - \log 1/n \Big).
\end{aligned} \tag{18}$$

where $\lambda > 0$ is a weighting coefficient. This term balances overall hidden representation editor utilization without forcing each token's distribution to be uniform. In practice, for numerical stability our work evaluate the log on $\max(\bar{p}_i, \varepsilon)$ with a small $\varepsilon$. The total objective becomes

$$\mathcal{L}_{\text{total}} \ = \ \mathcal{L}_{\text{main}} \ + \ \mathcal{L}_{\text{LB}}. \tag{19}$$

Table 9: Datasets and metrics used.

| task | training set | test set | metrics |
|------|-------------|----------|---------|
| Knowledge Reasoning | Commonsense-170K | BoolQ, PIQA, SIQA HellaSwag, WinoGrande ARC-e/c, OBQA | Accuracy |
| Mathematical Reasoning | Math-10K | MultiArith, GSM8k, SVAMP, MAWPS, AddSub, AQuA, SingleEq | Accuracy |
| GLUE | MNLI, SST-2, MRPC, CoLA, QNLI, QQP, RTE, STS-B train | MNLI, SST-2, MRPC, CoLA, QNLI, QQP, RTE, STS-B test | Matthews Correlation F1, Accuracy Pearson, Spearmanr |
| Conditional Text Generation | E2E-Challenge train | E2E-Challenge test | BLEU, NIST, METEOR, ROUGE–L, CIDEr |
| Code Synthesis | HumanEval, MBPP test (90%) | HumanEval, MBPP test (10%) | Pass@1 Rate |
| Knowledge Completion | WikiFact train | WikiFact test | Accuracy |
| Closed-Book QA | ScienceQA train | ScienceQA test | Accuracy |
| Symbolic Reasoning | CoinFlip train | CoinFlip test | Accuracy |
| Instruction Following | WizardLM | MT-Bench | First Turn Score |

## A.5 DATASET

Our work extensively evaluate the proposed TARE method on a suite covering nine capability categories: conditional text generation, code synthesis, knowledge completion, symbolic reasoning, closed-book QA, commonsense reasoning, mathematical reasoning, instruction following, and the GLUE benchmark. The datasets and metrics for each task are as follows (see Table 9):

- **Knowledge Reasoning** trains on Commonsense-170K Hu et al. (2023a) and tests on BoolQ Clark et al. (2019), PIQA Bisk et al. (2019), SIQA Sap et al. (2019), HellaSwag Zellers et al. (2019), WinoGrande Sakaguchi et al. (2019), ARC-e/c Clark et al. (2018), and OBQA Mihaylov et al. (2018), reporting accuracy;

- **Mathematical Reasoning** trains on Math-10K Hu et al. (2023b) and tests on MultiArith Roy & Roth (2015), GSM8k Cobbe et al. (2021), SVAMP Patel et al. (2021), MAWPS Koncel-Kedziorski et al. (2016), AddSub Hosseini et al. (2014), AQuA Ling et al. (2017), and SingleEq Koncel-Kedziorski et al. (2015), reporting accuracy;

- **GLUE** Wang et al. (2019) uses the official train/test splits, evaluating MNLI, SST-2, MRPC, CoLA, QNLI, QQP, RTE, and STS-B with the standard metrics (Matthews correlation, F1, Accuracy, Pearson, and Spearman).

- **Conditional Text Generation** uses the E2E-Challenge Novikova et al. (2017) train/test split and reports BLEU, NIST, METEOR, ROUGE-L, and CIDEr;

- **Code Synthesis** is uses HumanEval Chen et al. (2021) and MBPP Austin et al. (2021), training on 90% of the datasets (the remaining 10% as test) and evaluating Pass@1 Rate on the official HumanEval/MBPP test sets;

- **Knowledge Completion** uses WikiFact Goodrich et al. (2019) with accuracy;

- **Closed-Book QA** uses ScienceQA Saikh et al. (2022) with accuracy;

- **Symbolic Reasoning** uses CoinFlip Wei et al. (2022) with accuracy;

- **Instruction Following** trains on WizardLM Xu et al. (2024) and tests on MT-Bench Zheng et al. (2023), reporting First Turn Score judged by GPT-4 OpenAI et al. (2023).

### A.6 BASELINE

The following state-of-the-art baselines are used to compare with our proposed TARE method.

- **LoRA** Hu et al. (2021): injects trainable low-rank matrices $\mathbf{AB}^{\top}$ into the updates of linear layers while keeping the original weights frozen; our work follow the authors' defaults with rank $r=32$ and scaling $\alpha=32$.

- **DoRA** Liu et al. (2024): decouples the adaptation of direction and radius in weight space, improving optimization stability while maintaining a low update rank.

- **MiLoRA** Wang et al. (2024a): performs SVD on each weight matrix, keeps the principal singular subspace frozen, and attaches LoRA-style low-rank adapters to the minor subspace; during fine-tuning only these adapters are trained.

- **LoReFT** Wu et al. (2024b): applies low-rank re-parameterization jointly across layers and transfers features between tasks via a gating mechanism; our work use the public configuration with rank 8.

- **RED** Wu et al. (2024a): edits hidden representations directly by learning per-feature scaling and bias, without introducing inference-time modules.

- **BitFit** Ben Zaken et al. (2021): tunes only the bias terms in Transformer layers (e.g., attention and feed-forward blocks) while keeping all other weights frozen; introduces virtually no inference-time overhead.

- **IA³** Liu et al. (2022): applies learned per-feature multiplicative gates to key/value and feed-forward activations, modulating channels without changing the backbone weights; requires no rank hyperparameters and adds negligible inference cost.

- **LIFT** Liu et al. (2025): proposes Low-rank Informed Sparse Fine-Tuning, where each weight matrix is first approximated by a rank-$r$ SVD and then only the top-$k$ largest-magnitude entries of this low-rank approximation (Principal Weights) are selected as trainable parameters; during SFT, gradients and optimizer states are stored only for these Principal Weights.

- **WeGeFT** Savadikar et al. (2025): learns to generate weight-aware low-rank residuals directly from the frozen pretrained weights using shared low-rank matrices $\phi$ and $\psi$, so that each selected layer is updated as $\hat{\mathbf{W}}_{\ell} = \mathbf{W}_{\ell}(\mathbf{I} + \phi\psi)$.

- **PiSSA** Meng et al. (2024): computes the singular value decomposition of each weight matrix and uses the top-$r$ singular values and singular vectors to initialize a low-rank adapter $\mathbf{AB}^{\top}$ while freezing the residual matrix.

- **Spectral Adapter** Zhang & Pilanci (2024): first performs SVD $W = USV^{\top}$ for each pretrained weight matrix and then fine-tunes only the top-$r$ singular directions, either by additive updates to the leading singular vectors (Spectral AdapterA) or by orthogonal rotations parameterized via Cayley transforms (Spectral AdapterR).

- **LoRA-GA** Wang et al. (2024b): proposes a gradient-aware initialization for LoRA, where the low-rank adapters $\mathbf{BA}$ are initialized from the singular vectors of the full gradient matrix so that the first-step update $\Delta(\eta\mathbf{BA})$ closely aligns with the full fine-tuning gradient $\Delta\mathbf{W}$.

- **LoRA-One** Zhang et al. (2025): computes the one-step full fine-tuning gradient and performs an SVD-based spectral initialization so that the LoRA adapters $\mathbf{AB}^{\top}$ are aligned with the top-$r$ singular subspaces of this gradient, already providing a close low-rank approximation to the target update.

We confirm that the experimental setup for the baselines, including backbone models, training processes, and data preprocessing, directly matches the conditions used for TARE. Any differences in training conditions between TARE and the baselines will be clearly explained to ensure a fair comparison. For reproducibility, detailed information about the datasets, model configurations, and hyperparameters are provided in A.5 and A.7.

Regarding the use of load-balancing auxiliary loss in TARE with a value of $\lambda = 0.02$, we clarify that none of the baseline methods use an equivalent loss function. This is because the baselines do not employ a routing mechanism in their architectures. This distinction is critical for evaluating the performance gains attributed to TARE's unique load-balancing approach, and it is addressed in the ablation study to provide clearer insights into the impact of this loss term on model performance.

### A.7    IMPLEMENTATION DETAIL

To cover both major Transformer branches, our work fine-tune and evaluate **TARE** on a decoder-only backbone (LLaMA-3-8B Dubey et al. (2024)) and an encoder backbone (RoBERTa-base/large Liu et al. (2019)). Unless otherwise noted, our work set the number of hidden representation editors to $n = 8$ and select $k = 3$ editors per token via the token-aware selector; the load-balancing auxiliary loss is used with coefficient $\lambda = 0.02$. All experiments are implemented in PyTorch 2.4.1 and run on NVIDIA A100 (80 GB) GPUs. Our work use AdamW with learning rate $9 \times 10^{-4}$ and batch size 32, and load base language models in `bfloat16` to reduce memory usage. The datasets and task-specific evaluation metrics are summarized in Table 9.

### A.8    GLUE

TARE attains the best GLUE averages with 84.8 on RoBERTa-base and 88.3 on RoBERTa-large (Table 10). It delivers the top scores on MRPC (91.5/92.3) and STS-B (90.6/92.1), remains competitive on MNLI and QNLI (base: 86.3/91.7; large: 90.0/94.6), and lags on a few tasks such as CoLA or QQP. On average, it improves over LoRA and Adapter-FFN by +0.1 points each on RoBERTa-base and by +0.2/+0.6 points on RoBERTa-large, while exceeding RED by +0.5 (base) and +0.4 (large). These gains come with modest parameter counts of 0.22M (base) and 0.59M (large), smaller than LoRA (0.29M/0.79M) and Adapter-FFN (0.30M/0.80M).

Table 10: GLUE results with RoBERTa base and large. Results for LoRA, Adapter-FFN, BitFit, IA[3] and RED follows Wu et al. (2024a).

| PEFT | Source | RoBERTa | Params.(M) | MNLI | SST-2 | MRPC | CoLA | QNLI | QQP | RTE | STS-B | Avg. |
|---|---|---|---|---|---|---|---|---|---|---|---|---|
| LoRA | ICLR 21 | base | 0.29 | 86.6 | 93.9 | 88.7 | 59.7 | **92.6** | **90.4** | 75.3 | 90.3 | 84.7 |
| Adapter-FFN | EMNLP 20 | base | 0.30 | **87.1** | 93.0 | 88.8 | 58.5 | 92.0 | 90.2 | 77.7 | 90.4 | 84.7 |
| BitFit | ACL 22 | base | 0.10 | 84.7 | **94.0** | 88.1 | 54.0 | 91.0 | 87.3 | 69.8 | 89.5 | 82.3 |
| IA[3] | NeurIPS 22 | base | 0.06 | 85.4 | 93.4 | 86.4 | 57.8 | 91.1 | 88.5 | 73.5 | 88.5 | 83.1 |
| RED | ACL 24 | base | 0.02 | 83.9 | 93.9 | 89.2 | **61.0** | 90.7 | 87.2 | **78.0** | 90.4 | 84.3 |
| **TARE (ours)** | This paper | base | 0.22 | 86.3 | 93.1 | **91.5** | 58.6 | 91.7 | 88.6 | 77.8 | **90.6** | **84.8** |
| LoRA | ICLR 21 | large | 0.79 | 90.2 | 96.0 | 89.8 | 65.5 | **94.7** | 90.7 | **86.3** | 91.7 | 88.1 |
| Adapter-FFN | EMNLP 20 | large | 0.80 | **90.3** | **96.1** | 90.5 | 64.4 | 94.3 | **91.3** | 84.8 | 90.2 | 87.7 |
| IA[3] | NeurIPS 22 | large | 0.15 | 90.1 | 94.5 | 87.1 | 63.2 | 93.9 | 89.3 | 85.3 | 91.5 | 86.9 |
| RED | ACL 24 | large | 0.05 | 89.5 | 96.0 | 90.3 | **68.1** | 93.5 | 88.8 | 86.2 | 91.3 | 87.9 |
| **TARE (ours)** | This paper | large | 0.59 | 90.0 | 94.5 | **92.3** | 67.9 | 94.6 | 89.4 | 85.5 | **92.1** | **88.3** |

### A.9    CODE SYNTHESIS

TARE delivers the strongest code synthesis, achieving Pass@1 Rate = 56.3 on HumanEval and 48.0 on MBPP in Table 11. It surpasses the best baseline on HumanEval by +12.5 points over LoReFT (43.8), and it leads MBPP by +2.0 points over LoRA/RED (46.0). TARE trains only 0.0392% of parameters and uses 20,008 MiB peak VRAM, which is markedly lower than LoRA (23,038 MiB) and DoRA (44,382 MiB), indicating superior accuracy with substantially lighter adaptation.

Table 11: Code Synthesis, Closed-Book QA and Symbolic Reasoning results with LLaMA-3-8B. HumanEval and MBPP report Pass@1 Rate(%). ScienceQA and CoinFlip report Accuracy(%).

| PEFT | Source | Params.(%) | VRAM(MiB) | HumanEval | MBPP | ScienceQA | CoinFlip |
|------|--------|-----------|-----------|-----------|------|-----------|----------|
| LoRA | ICLR 21 | 0.7002 | 23 038 | 31.3 | 46.0 | 92.6 | 50.8 |
| DoRA | ICML 24 | 0.7098 | 44 382 | 25.0 | 42.0 | 92.0 | 55.8 |
| MiLoRA | NAACL 25 | 0.7002 | 23 478 | 37.5 | 40.0 | 92.9 | 57.1 |
| LoReFT | NeurIPS 24 | 0.0260 | 20 116 | 43.8 | 42.0 | 92.4 | 53.5 |
| RED | ACL 24 | 0.0033 | 18 762 | 25.0 | 46.0 | 93.4 | 50.5 |
| **TARE (ours)** | This paper | 0.0392 | 20 008 | **56.3** | **48.0** | **94.5** | **57.1** |

## A.10 KNOWLEDGE COMPLETION

TARE achieves the highest average accuracy of 67.0 on WikiFact (Table 12). It also attains the best score on four of five relations—jurisdiction 86.0, country 69.0, capital 55.0, and continent 86.0—and ties for the top on capital_of with 41.0. The average gain is +8.0 over LoRA and DoRA, +7.0 over MiLoRA, +4.0 over RED, and +2.0 over LoReFT, while training only 0.0392% of parameters. Peak memory is 16,512,MiB, which is close to the lowest among baselines.

Table 12: Knowledge Completion results with LLaMA-3-8B across five relation domains (jurisdiction, country, capital, capital_of, continent). Entries report Accuracy(%).

| PEFT | Source | Params.(%) | VRAM(MiB) | jurisdiction | country | capital | capital_of | continent | Avg. |
|------|--------|-----------|-----------|--------------|---------|---------|------------|-----------|------|
| LoRA | ICLR 21 | 0.7002 | 17 676 | 77.0 | 58.0 | 40.0 | 39.0 | 82.0 | 59.0 |
| DoRA | ICML 24 | 0.7098 | 36 712 | 75.0 | 56.0 | 39.0 | 41.0 | 83.0 | 59.0 |
| MiLoRA | NAACL 25 | 0.7002 | 18 710 | 75.0 | 66.0 | 39.0 | 39.0 | 80.0 | 60.0 |
| LoReFT | NeurIPS 24 | 0.0260 | 18 492 | 83.0 | 65.0 | 51.0 | 39.0 | 85.0 | 65.0 |
| RED | ACL 24 | 0.0033 | 16 352 | 82.0 | 62.0 | 46.0 | 40.0 | 83.0 | 63.0 |
| **TARE (ours)** | This paper | 0.0392 | 16 512 | **86.0** | **69.0** | **55.0** | **41.0** | **86.0** | **67.0** |

## A.11 CLOSED-BOOK QA AND SYMBOLIC REASONING

On Closed-Book QA and Symbolic Reasoning, TARE attains 94.5 on ScienceQA and 57.1 on Coin-Flip(Table 11). It does so while tuning only 0.0392% of parameters and using about 20 GiB VRAM. Compared with strong baselines, TARE is +1.1 points over RED on ScienceQA and +6.3 over LoRA on CoinFlip, and it matches the best CoinFlip score of MiLoRA. We attribute these gains to token-aware diagonal editing, which lets the model apply per-token, per-dimension adjustments that sharpen factual recall (ScienceQA) and stabilize discrete rule following (CoinFlip) without adding inference overhead.

## A.12 ABLATION STUDY

Comparison of the full TARE (scaling plus bias) against variants that remove scaling or bias. Entries report accuracy on seven math reasoning datasets and the average. Params.(%) denotes the percentage of trainable parameters. VRAM(MiB) denotes peak GPU memory. The full model attains the highest average score of 76.7 with 0.0392% trainable parameters. Removing either component degrades performance, and the scaling-only variant (56.4) outperforms the bias-only variant (50.5).

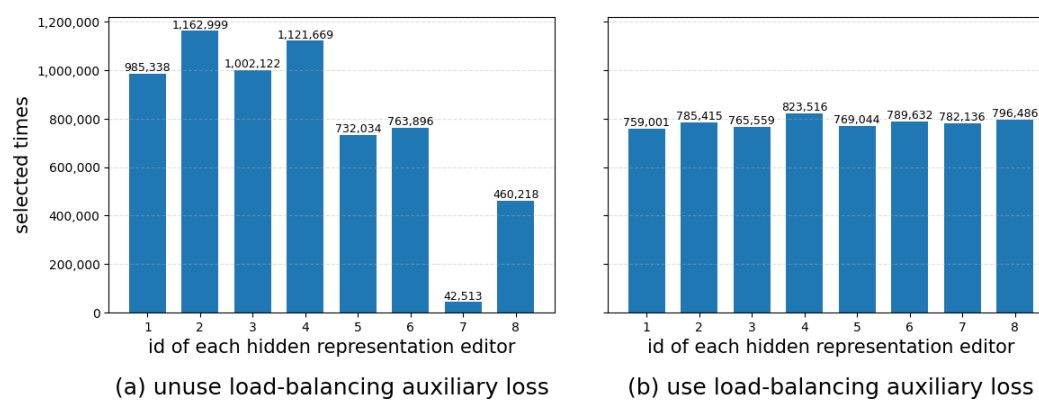

Figure 3: Effect of load balancing on editor utilization.

Table 13: Component ablation of TARE on LLaMA-3-8B.

| PEFT | Params.(%) | VRAM(MiB) | MultiArith | GSM8k | SVAMP | MAWPS | AddSub | AQuA | SingleEq | Avg. |
|---|---|---|---|---|---|---|---|---|---|---|
| **TARE (ours)** | 0.0392 | 20 900 | **95.8** | **57.3** | **72.9** | **86.1** | **90.9** | **41.4** | **92.1** | **76.7** |
| **TARE (w/o scaling)** | 0.0261 | 19 348 | 43.7 | 28.4 | 56.1 | 52.9 | 71.4 | 31.6 | 69.5 | 50.5 |
| **TARE (w/o bias)** | 0.0261 | 19 112 | 60.5 | 33.1 | 56.0 | 54.6 | 81.3 | 32.1 | 77.2 | 56.4 |

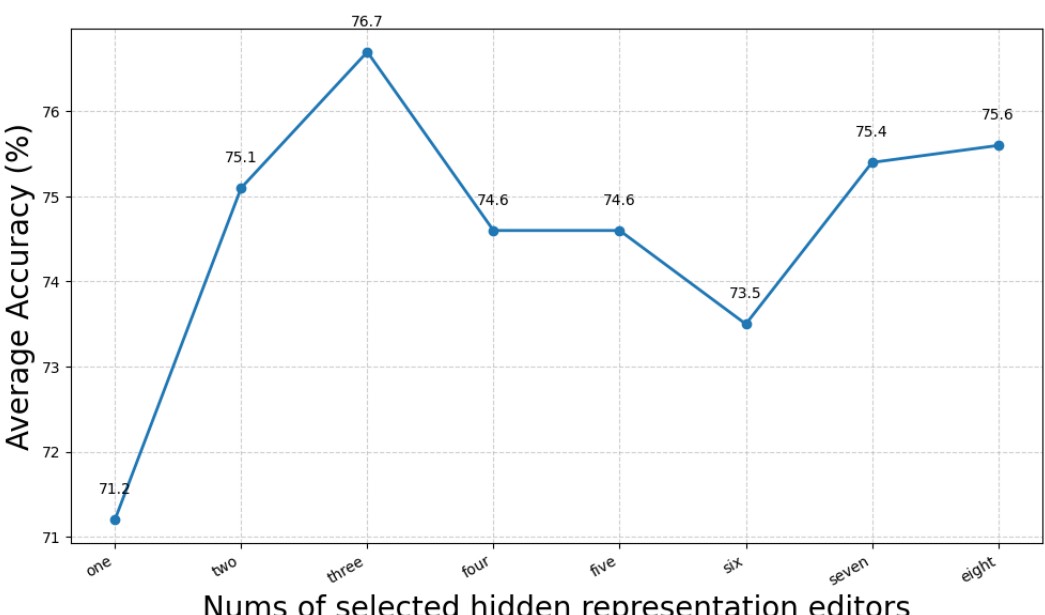

Figure 4: Sensitivity to Number of Selected Editors. Average accuracy on seven math-reasoning datasets (trained on Math-10K) as our work vary the count of token-wise selected hidden representation editors from one to eight. Accuracy rises sharply from one to three and peaks at three (76.7%). Larger selections show diminishing returns and fluctuate within 73.5–75.6%.

Effect of load-balancing auxiliary loss (n=8, k=3). With the loss, TARE attains a higher average accuracy (76.7 vs. 75.8) while keeping the trainable ratio fixed at 0.0392% and VRAM nearly unchanged. Improvements are seen on MultiArith (+2.0), GSM8k (+1.0), SVAMP (+0.5), AddSub

(+2.0), and AQuA (+1.6), with small changes on MAWPS (–0.5) and SingleEq (–0.6). This indicates that load balancing provides consistent gains without additional parameter or memory cost.

Table 14: Load-balancing ablation of TARE on LLaMA-3-8B.

| PEFT | Params.(%) | VRAM(MiB) | MultiArith | GSM8k | SVAMP | MAWPS | AddSub | AQuA | SingleEq | Avg. |
|---|---|---|---|---|---|---|---|---|---|---|
| **TARE (ours)** | 0.0392 | 20 900 | **95.8** | **57.3** | **72.9** | 86.1 | **90.9** | **41.4** | 92.1 | **76.7** |
| **TARE (w/o lb loss)** | 0.0392 | 20 842 | 93.8 | 56.3 | 72.4 | 86.6 | 88.9 | 39.8 | 92.7 | 75.8 |

## A.13 SENSITIVITY ANALYSIS

Table 15: Hyperparameter Sensitivity Analysis of k (selected nums of editors) on TARE.

| PEFT | Params.(%) | VRAM(MiB) | MultiArith | GSM8k | SVAMP | MAWPS | AddSub | AQuA | SingleEq | Avg. |
|---|---|---|---|---|---|---|---|---|---|---|
| **TARE (k=1)** | 0.0392 | 20 196 | 92.3 | 47.9 | 66.0 | 84.9 | 88.4 | 28.0 | 91.1 | 71.2 |
| **TARE (k=2)** | 0.0392 | 20 548 | 93.2 | 56.9 | 71.7 | 83.6 | 89.1 | 40.2 | 90.9 | 75.1 |
| **TARE (k=3)** | 0.0392 | 20 900 | **95.8** | 57.3 | 72.9 | 86.1 | **90.9** | 41.4 | 92.1 | **76.7** |
| **TARE (k=4)** | 0.0392 | 21 274 | 92.5 | 56.4 | 71.5 | 82.4 | 87.6 | 39.4 | 92.7 | 74.6 |
| **TARE (k=5)** | 0.0392 | 21 652 | 93.3 | 57.2 | 71.3 | 85.7 | 90.6 | 31.1 | 92.7 | 74.6 |
| **TARE (k=6)** | 0.0392 | 22 074 | 93.5 | 55.2 | **73.6** | 85.7 | 88.1 | 36.2 | **93.5** | 75.1 |
| **TARE (k=7)** | 0.0392 | 22 412 | 95.5 | **62.2** | 70.7 | 87.0 | 89.6 | 29.6 | 93.3 | 75.4 |
| **TARE (k=8)** | 0.0392 | 22 784 | 91.7 | 56.9 | 70.1 | **87.4** | 88.6 | **43.0** | 91.9 | 75.6 |

Table 16: Hyperparameter Sensitivity Analysis of n (total nums of editors) on TARE.

| PEFT | Params.(%) | VRAM(MiB) | MultiArith | GSM8k | SVAMP | MAWPS | AddSub | AQuA | SingleEq | Avg. |
|---|---|---|---|---|---|---|---|---|---|---|
| **TARE (n=6)** | 0.0294 | 20 892 | 92.8 | 60.4 | 73.0 | **88.1** | 89.1 | 38.2 | **93.7** | 76.5 |
| **TARE (n=8)** | 0.0392 | 20 900 | **95.8** | 57.3 | 72.9 | 86.1 | **90.9** | 41.4 | 92.1 | **76.7** |
| **TARE (n=10)** | 0.0489 | 20 904 | 93.7 | **62.1** | **74.1** | 87.7 | 87.1 | 35.8 | 91.9 | 76.1 |

