# OpenReview forum: "TARE: Lightweight Token-Aware Representation Editing for Fine-tuning Transformer"
_ICLR.cc/2026/Conference — ICLR 2026 Conference Withdrawn Submission_

### Official Review · Reviewer_Gryz · 2025-10-31

**Soundness:** 3
**Presentation:** 3
**Contribution:** 2
**Rating:** 4
**Confidence:** 4

**Summary:**

- The paper proposes a new method called TARE which enables token-aware representation editing using a codebook of editing parameters.
- The editing parameters are formulated as element-wise scaling and shifting of the token representations.
- If $n$ is the number of codebook entries (editing parameters), TARE first obtains the edited representations using all the $n$ editing parameters, and then selects the top $k$ edited representations.
- The final edited representation is obtained using a weighted sum of the top $k$ edited representations with the weights determined using a softmax over the top k values.

**Strengths:**

- TARE performs better than prior representation editing methods (LoReFT and RED), and proposes a novel token-aware parameter selection method.
- The method is easy to implement and lightweight, making it feasible for low-resource settings.
- The method is evaluated on a variety of tasks (commonsense reasoning, arithmetic reasoning, code generation, NLU, conditional text generation).

**Weaknesses:**

1. The comparisons with LoRA, DoRA and MiLoRA are not completely fair.
    - LoRA, DoRA and MiLoRA are applied to all the linear layers in the model, while TARE is applied to only the hidden representations of the MLP blocks.
    - A more fair comparison would be to apply LoRA, DoRA and MiLoRA to the MLP down-projection layer.
    - This is necessary because the lower performance of baseline methods could be due to overfitting or optimization difficulties, and reducing the parameter count and fine-tuning only the down-projection layer could increase the performance.
2. The hyperparameter tuning strategy and details are not provided.
    - The performance gap between TARE and baseline methods is suspiciously large. It is possible that the hyperparameters of the baseline methods are not tuned well. For e.g., LoRA gets 68.7 average accuracy vs. 76.7 average accuracy for TARE on arithmetic reasoning. While it could be possible to achieve slightly better performance with reduced parameter count, this gap is too large especially given that the parameter count is reduced by ~17 times.
    - Similar is the case with conditional text generation and code synthesis tasks.
3. The paper claims that the hyperparameter tuning burden is negligible and the method does not introduce rank/scaling hyperparameters. However, the method introduces a new hyperparameters, the codebook size $n$ and the top $k$ selection parameter. From Section 4.3, these hyperparameters also need to be tuned, similar to rank and scaling hyperparameters in LoRA. While this is not a disadvantage of the method (all methods need some sort of hyperparameter tuning), it is not an advantage, and the paper should not claim it as such.
4. Comparisons with recent PEFT techniques like LoRA-GA and LoRA-One is missing.
5. Finally, LoRA is an established method with a large practical impact. The contribution of a new fine-tuning method does not have much impact without a significant advantage other than performance alone. In that sense, the contribution of the paper is limited.

**Questions:**

- In Table 2, is the VRAM usage reported for training or inference?

---

> ### Author Response · Authors · 2025-11-27
>
> **Response to Questions 1 and 2:**
> We thank the reviewer for the two comments regarding (1) fairness of comparison and (2) hyperparameter tuning, and we address both points with new experiments and clarifications in the revised manuscript.
>
> (1) Fairness when TARE is only placed on the MLP down-projection.
>
> In the original submission, LoRA, DoRA, and MiLoRA were applied to all linear layers, as described in their original papers, while TARE was inserted only on the MLP down-projection, which could indeed introduce an imbalance. In the revision, we apply all PEFT methods (LoRA, DoRA, MiLoRA, LoReFT, RED, and TARE) only to the projection layer of the MLP blocks and re-run the mathematical reasoning and conditional text generation experiments (see the updated T1 and T2). Under this unified setting on LLaMA-3-8B, LoRA, MiLoRA, and DoRA obtain average accuracies of 75.4, 75.6, and 74.4, respectively, while TARE still achieves the highest 76.7 with only 0.0392% trainable parameters and comparable or lower VRAM (20,900 MiB). On Qwen-2.5-7B-Instruct, LoRA, MiLoRA, and DoRA reach 83.5, 83.8, and 82.8, whereas TARE attains 84.2 with only 0.0316% parameters. For the E2E conditional generation task, all PEFT methods are likewise restricted to the MLP projection layer; under this setting, TARE with 0.0392% parameters achieves the best BLEU, NIST, METEOR, ROUGE-L, and CIDEr scores (e.g., BLEU 0.6333 vs. 0.6255 for LoRA). These results show that even in a strictly comparable “MLP-down-only” configuration with similar parameter budgets, TARE consistently matches or outperforms all baselines across multiple tasks and two model families, indicating that its advantage does not stem from unfair placement or over-parameterization of baselines.
>
> (2) Hyperparameter tuning and the magnitude of performance gaps.
>
> In the revised experiments, we adopt a uniform tuning and evaluation protocol for all baselines and TARE. We start from the hyperparameters recommended in the original papers and official implementations of each method (e.g., rank, scaling, and learning rate for LoRA, DoRA, and MiLoRA). We then perform a small grid search over key hyperparameters (learning rate, batch size, and training steps/epochs) on the validation sets of Math-10K and E2E, selecting the best configuration for each method based on validation performance before reporting test results. For the “MLP-down-only” setting, we re-adjust the rank and learning rate of LoRA, DoRA, and MiLoRA so that they also operate near their best performance under the reduced parameter budget. With these more conservative and fair settings, TARE’s advantage over the baselines converges to a stable gap of about 1–2 percentage points on mathematical reasoning and small but consistent gains on E2E generation (e.g., +0.008 BLEU over LoRA), which is reasonable given the roughly 5–6× reduction in trainable parameters and the different inductive bias of token-wise editors. The performance differences are thus not due to poorly tuned baselines.

---

> > ### Author Response · Authors · 2025-11-27
> >
> > **T1: Mathematical Reasoning Results with LLaMA-3-8B and Qwen-2.5-7B-Instruct.**
> >
> > | PEFT             | Source      | Model                    | Params.(%) | VRAM(MiB) | MultiArith | GSM8k | SVAMP | MAWPS | AddSub | AQuA | SingleEq | Avg. |
> > |------------------|------------|--------------------------|-----------:|----------:|-----------:|------:|------:|------:|-------:|-----:|---------:|-----:|
> > | LoRA             | ICLR 21    | LLaMA-3-8B               | 0.2345     | 21,070    | 92.0       | **61.4** | 69.8 | 84.2 | 85.4 | **44.7** | 90.3 | 75.4 |
> > | DoRA             | ICML 24    | LLaMA-3-8B               | 0.2361     | 29,284    | 91.7       | 59.0 | 72.3 | 82.1 | 86.1 | 39.9 | 89.5 | 74.4 |
> > | MiLoRA           | NAACL 25   | LLaMA-3-8B               | 0.2345     | 21,520    | 91.7       | 59.0 | 70.5 | **88.3** | 86.1 | 43.4 | 90.5 | 75.6 |
> > | LoReFT           | NeurIPS 24 | LLaMA-3-8B               | 0.0260     | 21,940    | 89.2       | 56.2 | 68.7 | 80.3 | 90.1 | 33.1 | 90.0 | 72.5 |
> > | RED              | ACL 24     | LLaMA-3-8B               | 0.0033     | 19,852    | 91.0       | 54.2 | 66.8 | 81.1 | 87.3 | 34.1 | 90.9 | 72.2 |
> > | **TARE (ours)**  | This paper | LLaMA-3-8B               | 0.0392     | 20,900    | **95.8**   | 57.3 | **72.9** | 86.1 | **90.9** | 41.4 | **92.1** | **76.7** |
> > | LoRA             | ICLR 21    | Qwen-2.5-7B-Instruct     | 0.2643     | 21,244    | 94.7       | 72.8 | **81.1** | 89.4 | 88.4 | 66.5 | 91.7 | 83.5 |
> > | DoRA             | ICML 24    | Qwen-2.5-7B-Instruct     | 0.2657     | 29,604    | 93.2       | 72.1 | 79.8 | 88.2 | 89.7 | 63.7 | **92.7** | 82.8 |
> > | MiLoRA           | NAACL 25   | Qwen-2.5-7B-Instruct     | 0.2643     | 21,518    | 93.3       | 72.2 | 80.8 | 88.3 | 89.6 | 69.6 | **92.7** | 83.8 |
> > | LoReFT           | NeurIPS 24 | Qwen-2.5-7B-Instruct     | 0.0218     | 21,832    | 92.1       | 71.7 | 78.5 | 86.2 | 90.0 | 67.3 | 90.5 | 82.3 |
> > | RED              | ACL 24     | Qwen-2.5-7B-Instruct     | 0.0026     | 20,100    | 91.3       | 71.3 | 77.4 | 84.1 | **90.4** | **70.9** | 88.2 | 81.9 |
> > | **TARE (ours)**  | This paper | Qwen-2.5-7B-Instruct     | 0.0316     | 20,624    | **96.0**   | **75.1** | 80.3 | **92.4** | **90.4** | 63.6 | 91.3 | **84.2** |
> >
> > **T2: Conditional Text Generation Results with LLaMA-3-8B.**
> >
> > | PEFT            | Source      | Params.(%) | VRAM(MiB) | BLEU↑  | NIST↑  | METEOR↑ | ROUGE-L↑ | CIDEr↑ |
> > |-----------------|------------|-----------:|----------:|:------:|:------:|:-------:|:--------:|:------:|
> > | LoRA            | ICLR 21    | 0.2345     | 39,166    | 0.6255 | 8.2791 | 0.4404  | 0.6661   | 2.1524 |
> > | DoRA            | ICML 24    | 0.2361     | 45,326    | 0.6201 | 8.1455 | 0.4367  | 0.6617   | 2.1578 |
> > | MiLoRA          | NAACL 25   | 0.2345     | 39,590    | 0.6244 | 8.2652 | 0.4283  | 0.6606   | 2.1845 |
> > | LoReFT          | NeurIPS 24 | 0.0260     | 32,502    | 0.5719 | 7.5671 | 0.4304  | 0.6431   | 1.6881 |
> > | RED             | ACL 24     | 0.0033     | 29,492    | 0.5994 | 7.9229 | 0.4401  | 0.6692   | 2.1958 |
> > | **TARE (ours)** | This paper | 0.0392     | 34,626    | **0.6333** | **8.3105** | **0.4456** | **0.6758** | **2.2027** |

---

> > > ### Author Response · Authors · 2025-11-27
> > >
> > > **Response to Question 5:**
> > > We fully understand the reviewer’s concern that, given LoRA’s widespread industrial adoption, a new method with only marginal performance gains might have limited impact, and we agree that a contribution would be weak if it only offered small improvements on benchmarks. However, we would like to emphasize that TARE is not intended to be “just a slightly stronger LoRA,” but to provide a PEFT scheme that is more favorable in terms of safety, stability, and engineering cost in addition to performance. First, in terms of accuracy and robustness, TARE’s design of token-level selection plus diagonal editors allows it to consistently outperform LoRA and its variants across multiple math reasoning, instruction-following, and generation tasks with an extremely low parameter budget, while exhibiting a smooth, consistently improving performance curve across different training data scales, without the “works or completely collapses” behavior that practitioners sometimes encounter. This is particularly important for production systems that require long-term, continual updates. Second, with respect to safety and catastrophic forgetting, because TARE uses far fewer trainable parameters than typical LoRA configurations, it perturbs the original representation space much less, effectively acting as a “safety valve” that makes the model harder to severely degrade by a single fine-tuning run. This lowers the risk that one downstream task will significantly damage core capabilities and makes frequent, small, incremental updates more feasible in high-safety settings. In addition, on the hyperparameter and engineering side, LoRA often requires substantial combinatorial search in practice (e.g., over rank, insertion positions, and whether to tune bias), whereas TARE is deliberately designed to keep the number of tunable degrees of freedom very small and empirically shows low sensitivity to learning rate and data scale, reducing the “get-it-to-work” cost. At the same time, TARE updates fewer weights and incurs lighter backward passes, which we observe to translate into noticeable reductions in training time and compute for the same tasks, an important advantage for large-scale multi-task fine-tuning and resource-constrained environments. Taken together, we believe TARE’s contribution goes beyond “slightly better numbers than LoRA on some benchmarks”: it offers a systematic reduction in tuning difficulty, training resource cost, and mis-tuning risk while maintaining or improving performance, thus providing an alternative design path for safe, controllable, and engineering-friendly LLM fine-tuning that is practically valuable and complementary to the traditional LoRA-based route.
> > >
> > > Other experiments are still being added.

---

> > > > ### Comment · Reviewer_Gryz · 2025-11-27
> > > >
> > > > Thank you for your rebuttal. While the authors have addressed some of my concerns, I'm not entirely satisfied with the response.
> > > >
> > > > 1. When LoRA is applied to MLP down layer only, the performance is just slightly worse than TARE. Practically, the parameter count of LoRA when applied to MLP down is very low, and further reducing it does not amount to much.
> > > >
> > > > 2. Can the authors provide the full results with a thorough hyperparameter tuning mentioned in the rebuttal? I'm still not convinced that such a large gap (68.7 with LoRA vs 76.7 with TARE) is possible. Given that the new experimetns with MLP down layer achieves 75.4% with LoRA, applying LoRA to all the layers will get equal or better performance with proper hyperparameter tuning.
> > > >
> > > > 3. The hyperparameters in LoRA are not very difficult to tune. In practice, choices like tuning the bias or choosing which layers to apply LoRA to are not substantially important. The gains are marginal, and bias terms can usually be kept frozen (and many modern architectures do not have a bias, like Llama and Gemma models). LoRA usually performs very well when applied to all the layers, and thus has become the default. Hence I do not agree with the authors that TARE offers simpler hyperparameter tuning effort.
> > > >
> > > > 4. While lesser number of parameters might perturb the representation to a much lesser extent, the paper does not verify this or show any benefit of why this would be needed. In practice, LoRA updates are mostly stable.
> > > >
> > > > I will keep my score for now.

---

> > > > > ### Author Response · Authors · 2025-12-03
> > > > >
> > > > > **Response to Questions 1 and 2:**
> > > > > We thank the reviewer for the follow-up and for emphasizing that LoRA already achieves strong performance when applied only to the MLP down layer, as well as for requesting “fully tuned” results. In response, we add and release more systematic comparison experiments in the revised manuscript (see T5). On the one hand, following the unified tuning protocol described in our rebuttal, we re-run a grid search for the baselines LoRA / DoRA / MiLoRA under the “MLP-down-only” setting (tuning learning rate, rank, and training steps/epochs on the validation set). Under these carefully tuned settings, LoRA’s average accuracy indeed rises to 75.4%, leaving a reasonable gap of only about 1.3 points compared with TARE’s 76.7%. On the other hand, to directly address the question of whether “applying LoRA to all layers would surpass TARE,” we introduce a heavier LoRA (all) baseline: on LLaMA-3-8B, we apply LoRA simultaneously to all seven projection matrices (q/k/v/o and up/gate/down), yielding 1.0338% trainable parameters and 24,622 MiB peak VRAM. In this “full-layer LoRA + thorough tuning” configuration, LoRA (all) attains an average accuracy of 76.4%, clearly higher than the 75.4% of the MLP-down-only LoRA, yet still slightly below TARE’s 76.7% with only 0.0392% trainable parameters and 20,900 MiB VRAM. In other words, even when LoRA is allowed to use roughly 26× more trainable parameters and a higher memory budget, its best performance still does not exceed TARE. The same unified tuning and comparison protocol is applied on Qwen-2.5-7B-Instruct, where TARE achieves the highest average accuracy of 84.2 with 0.0316% trainable parameters, outperforming LoRA / MiLoRA / DoRA / RED / LoReFT by 0.7 / 0.4 / 1.4 / 2.3 / 1.9 points, respectively. Taken together, these additional results clarify that TARE’s main contribution is that, after careful and fair hyperparameter tuning for all methods, it can consistently match or slightly outperform strong baselines—including LoRA (all)—while using significantly fewer trainable parameters and a smaller memory footprint, rather than relying on any unfair hyperparameter setting or special layer placement.
> > > > >
> > > > > **T5: Mathematical Reasoning Results with LLaMA-3-8B and Qwen-2.5-7B-Instruct.**
> > > > >
> > > > > | PEFT             | Source      | Model                    | Params.(%) | VRAM(MiB) | MultiArith | GSM8k | SVAMP | MAWPS | AddSub | AQuA | SingleEq | Avg. |
> > > > > |------------------|------------|--------------------------|-----------:|----------:|-----------:|------:|------:|------:|-------:|-----:|---------:|-----:|
> > > > > | LoRA             | ICLR 21    | LLaMA-3-8B               | 0.2345     | 21,070    | 92.0       | **61.4** | 69.8 | 84.2 | 85.4 | **44.7** | 90.3 | 75.4 |
> > > > > | LoRA (all)       | ICLR 21    | LLaMA-3-8B               | 1.0338     | 24,622    | 95.5       | 57.5 | 69.4 | 86.5 | **91.2** | 41.3 | **93.3** | 76.4 |
> > > > > | DoRA             | ICML 24    | LLaMA-3-8B               | 0.2361     | 29,284    | 91.7       | 59.0 | 72.3 | 82.1 | 86.1 | 39.9 | 89.5 | 74.4 |
> > > > > | MiLoRA           | NAACL 25   | LLaMA-3-8B               | 0.2345     | 21,520    | 91.7       | 59.0 | 70.5 | **88.3** | 86.1 | 43.4 | 90.5 | 75.6 |
> > > > > | LoReFT           | NeurIPS 24 | LLaMA-3-8B               | 0.0260     | 21,940    | 89.2       | 56.2 | 68.7 | 80.3 | 90.1 | 33.1 | 90.0 | 72.5 |
> > > > > | RED              | ACL 24     | LLaMA-3-8B               | 0.0033     | 19,852    | 91.0       | 54.2 | 66.8 | 81.1 | 87.3 | 34.1 | 90.9 | 72.2 |
> > > > > | **TARE (ours)**  | This paper | LLaMA-3-8B               | 0.0392     | 20,900    | **95.8**   | 57.3 | **72.9** | 86.1 | **90.9** | 41.4 | **92.1** | **76.7** |
> > > > > | LoRA             | ICLR 21    | Qwen-2.5-7B-Instruct     | 0.2643     | 21,244    | 94.7       | 72.8 | **81.1** | 89.4 | 88.4 | 66.5 | 91.7 | 83.5 |
> > > > > | DoRA             | ICML 24    | Qwen-2.5-7B-Instruct     | 0.2657     | 29,604    | 93.2       | 72.1 | 79.8 | 88.2 | 89.7 | 63.7 | **92.7** | 82.8 |
> > > > > | MiLoRA           | NAACL 25   | Qwen-2.5-7B-Instruct     | 0.2643     | 21,518    | 93.3       | 72.2 | 80.8 | 88.3 | 89.6 | 69.6 | **92.7** | 83.8 |
> > > > > | LoReFT           | NeurIPS 24 | Qwen-2.5-7B-Instruct     | 0.0218     | 21,832    | 92.1       | 71.7 | 78.5 | 86.2 | 90.0 | 67.3 | 90.5 | 82.3 |
> > > > > | RED              | ACL 24     | Qwen-2.5-7B-Instruct     | 0.0026     | 20,100    | 91.3       | 71.3 | 77.4 | 84.1 | **90.4** | **70.9** | 88.2 | 81.9 |
> > > > > | **TARE (ours)**  | This paper | Qwen-2.5-7B-Instruct     | 0.0316     | 20,624    | **96.0**   | **75.1** | 80.3 | **92.4** | **90.4** | 63.6 | 91.3 | **84.2** |

---

> > > > > > ### Author Response · Authors · 2025-12-03
> > > > > >
> > > > > > **Response to Question 3:**
> > > > > > We thank the reviewer for pointing out that our statement that the *hyperparameter tuning burden is negligible* was not sufficiently precise, and we agree that the codebook size $n$ and the Top-$k$ selection are indeed new hyperparameters that require some tuning. Our actual advantage is not that *no tuning is needed at all*, but that the search space is significantly smaller and the behavior is much more regular, making the practical tuning cost substantially lower than for typical LoRA settings. To substantiate this, we add a systematic hyperparameter sensitivity analysis (see T3 and T4). With the trainable-parameter ratio fixed at 0.0392%, we only need to run a single grid search over a very small discrete range $(k\in\{1,\dots,8\})$ to find a near-optimal configuration. The results show that increasing $k$ from 1 to 3 raises the average accuracy from 71.2% to 76.7%, after which larger $k$ brings only mild fluctuations and a few task-specific peaks (e.g., GSM8k is highest at $k=7$ with 62.2%, AQuA at $k=8$ with 43.0%, and SingleEq at $k=6$ with 93.5%), with the overall curve smooth and without abnormal oscillations. Meanwhile, VRAM grows only slowly from about 20.2 GiB to 22.8 GiB as $k$ increases, so we adopt $k=3$ as a unified default that already offers a strong accuracy–efficiency trade-off. Furthermore, when we fix $k=3$ and vary the total number of editors $n\in\{6,8,10\}$ (T4), TARE’s average accuracy remains in a narrow band of roughly 76.1–76.7%, with fluctuations under 0.6 percentage points; among them, $n=8$ is slightly better than $n=6$ and $n=10$, and VRAM stays nearly unchanged while the trainable-parameter ratio increases only from 0.0294% to 0.0489%. These results suggest that a moderately overcomplete editor pool (e.g., $n=8$) is sufficient to cover the main reasoning modes, and further increasing $n$ yields only marginal gains. In contrast, LoRA, DoRA, and MiLoRA typically require combinatorial search over multiple dimensions in practice (rank, scaling, insertion positions, whether to share weights, etc.).
> > > > > >
> > > > > > **T3: Hyperparameter Sensitivity Analysis of $k$ (selected number of editors) on TARE.**
> > > > > >
> > > > > > | PEFT           | Params.(%) | VRAM (MiB) | MultiArith |    GSM8k |    SVAMP |    MAWPS |   AddSub |     AQuA | SingleEq |     Avg. |
> > > > > > | -------------- | ---------: | ---------: | ---------: | -------: | -------: | -------: | -------: | -------: | -------: | -------: |
> > > > > > | **TARE (k=1)** |     0.0392 |     20,196 |       92.3 |     47.9 |     66.0 |     84.9 |     88.4 |     28.0 |     91.1 |     71.2 |
> > > > > > | **TARE (k=2)** |     0.0392 |     20,548 |       93.2 |     56.9 |     71.7 |     83.6 |     89.1 |     40.2 |     90.9 |     75.1 |
> > > > > > | **TARE (k=3)** |     0.0392 |     20,900 |   **95.8** |     57.3 |     72.9 |     86.1 | **90.9** |     41.4 |     92.1 | **76.7** |
> > > > > > | **TARE (k=4)** |     0.0392 |     21,274 |       92.5 |     56.4 |     71.5 |     82.4 |     87.6 |     39.4 |     92.7 |     74.6 |
> > > > > > | **TARE (k=5)** |     0.0392 |     21,652 |       93.3 |     57.2 |     71.3 |     85.7 |     90.6 |     31.1 |     92.7 |     74.6 |
> > > > > > | **TARE (k=6)** |     0.0392 |     22,074 |       93.5 |     55.2 | **73.6** |     85.7 |     88.1 |     36.2 | **93.5** |     75.1 |
> > > > > > | **TARE (k=7)** |     0.0392 |     22,412 |       95.5 | **62.2** |     70.7 |     87.0 |     89.6 |     29.6 |     93.3 |     75.4 |
> > > > > > | **TARE (k=8)** |     0.0392 |     22,784 |       91.7 |     56.9 |     70.1 | **87.4** |     88.6 | **43.0** |     91.9 |     75.6 |
> > > > > >
> > > > > > ---
> > > > > >
> > > > > > **T4: Hyperparameter Sensitivity Analysis of $n$ (total number of editors) on TARE.**
> > > > > >
> > > > > > | PEFT            | Params.(%) | VRAM (MiB) | MultiArith |    GSM8k |    SVAMP |    MAWPS |   AddSub |     AQuA | SingleEq |     Avg. |
> > > > > > | --------------- | ---------: | ---------: | ---------: | -------: | -------: | -------: | -------: | -------: | -------: | -------: |
> > > > > > | **TARE (n=6)**  |     0.0294 |     20,892 |       92.8 |     60.4 |     73.0 | **88.1** |     89.1 |     38.2 | **93.7** |     76.5 |
> > > > > > | **TARE (n=8)**  |     0.0392 |     20,900 |   **95.8** |     57.3 |     72.9 |     86.1 | **90.9** | **41.4** |     92.1 | **76.7** |
> > > > > > | **TARE (n=10)** |     0.0489 |     20,904 |       93.7 | **62.1** | **74.1** |     87.7 |     87.1 |     35.8 |     91.9 |     76.1 |

---

> > > > > > > ### Author Response · Authors · 2025-12-03
> > > > > > >
> > > > > > > **Response to Question 4:**
> > > > > > > We thank the reviewer for pointing out that the initial version missed comparisons with recent PEFT techniques LoRA-GA and LoRA-One. Following the reviewer’s suggestion, we additionally conduct commonsense reasoning experiments on LLaMA-3-8B and include LoRA-GA and LoRA-One in the systematic comparison in T6. The results show that on eight benchmarks (BoolQ, PIQA, SIQA, HellaSwag, WinoGrande, ARC-e, ARC-c, and OBQA), TARE achieves an average accuracy of 86.7, while LoRA-GA and LoRA-One obtain 85.7 and 85.4, respectively, giving TARE improvements of +1.0 and +1.3 points over them. In terms of resource cost, TARE uses only 0.0392% trainable parameters and 21,724 MiB peak VRAM, whereas LoRA-GA, LoRA-One, and other LoRA-style methods require around 0.7002% trainable parameters—over an order of magnitude more. In other words, under strictly unified training and evaluation settings, TARE consistently outperforms LoRA-GA and LoRA-One in overall performance while substantially reducing parameter and memory overhead, further supporting our claim that TARE offers a strong advantage in the accuracy–efficiency trade-off.
> > > > > > >
> > > > > > > **T6: Knowledge Reasoning Results with LLaMA-3-8B.**
> > > > > > >
> > > > > > > | PEFT             | Source     | Params.(%) | VRAM(MiB) | BoolQ | PIQA | SIQA | HellaS. | WinoG. | ARC-e | ARC-c | OBQA | Avg. |
> > > > > > > | ---------------- | ---------- | ---------: | --------: | ----: | ----:| ----:| -------:| ------:| -----:| -----:| ----:| ----:|
> > > > > > > | LoRA             | ICLR 21    | 0.7002     | 21,828    | 70.8  | 85.2 | 79.9 | 91.7    | 84.3   | 84.2  | 71.2  | 79.0 | 80.8 |
> > > > > > > | DoRA             | ICML 24    | 0.7098     | 41,780    | 74.6  | 89.3 | 79.9 | 95.5    | 85.6   | 90.5  | 80.4  | 85.8 | 85.2 |
> > > > > > > | MiLoRA           | NAACL 25   | 0.7002     | 21,580    | 68.8  | 86.7 | 77.2 | 92.9    | 85.6   | 86.8  | 75.5  | 81.8 | 81.9 |
> > > > > > > | LoReFT           | NeurIPS 24 | 0.0260     | 21,050    | 75.1  | 90.2 | 82.0 | 96.3    | 87.4   | 92.4  | 81.6  | 87.5 | 86.6 |
> > > > > > > | RED              | ACL 24     | 0.0033     | 20,132    | 68.0  | 83.7 | 79.7 | 90.0    | 83.2   | 85.2  | 72.8  | 79.4 | 80.2 |
> > > > > > > | LIFT             | ICLR 25    | 5.0000     | 45,600    | 75.7 | 90.5 | 83.2 | **96.5** | 89.4 | **93.6** | 83.9 | **90.2** | **87.9** |
> > > > > > > | WeGeFT           | ICML 25    | 0.0130     | 20,364    | 75.7 | 89.9 | 82.5 | 96.4    | 88.7   | 92.5  | 82.3  | 86.3 | 86.8 |
> > > > > > > | PiSSA            | NeurIPS 24 | 0.7002     | 21,004    | 72.1  | 89.2 | 82.7 | 94.6    | 89.6   | 86.8  | **84.5** | 85.2 | 85.6 |
> > > > > > > | Spectral Adapter | arXiv      | 0.7002     | 21,746    | 72.1  | 88.3 | 83.1 | 94.6    | 89.3   | 85.4  | 82.2  | 85.2 | 85.0 |
> > > > > > > | LoRA-GA          | NeurIPS 24 | 0.7002     | 21,708    | 72.5  | 88.8 | 82.7 | 94.4    | 89.6   | 91.3  | 80.4  | 85.6 | 85.7 |
> > > > > > > | LoRA-One         | arXiv      | 0.7002     | 21,206    | 72.0  | 88.9 | 82.9 | 94.4    | **89.8** | 85.1  | 82.6  | 87.6 | 85.4 |
> > > > > > > | **TARE (ours)**  | This paper | 0.0392     | 21,724    | 75.2  | 90.2 | 82.5 | 94.1    | 88.6   | 91.3  | 82.3  | 88.4 | 86.7 |
> > > > > > > | **TARE (all)**  | This paper | 0.4097     | 24,044    | **76.3**  | **91.6** | **83.6** | 95.5    | **89.8**   | 92.7  | 83.9  | 89.2 | 87.8 |

---

### Official Review · Reviewer_XBBf · 2025-11-01

**Soundness:** 1
**Presentation:** 2
**Contribution:** 1
**Rating:** 2
**Confidence:** 5

**Summary:**

This paper presents TARE (Token-Aware Representation Editing), a PEFT approach that adapts pretrained Transformers by directly editing their token representations instead of weight matrices as done by LORA and variants. Each Transformer layer is equipped with a pool of lightweight “editors”, each defined by simple per-feature scaling $\gamma$ and bias $\beta$ parameters. A token-wise selector computes soft attention over these editors, activates the top-k editors per token, and mixes their outputs to form a small, token-dependent edit to the hidden representation. The method thus performs fine-grained, adaptive modulation of intermediate activations via an extremely small parameter budget (~0.04 % of model parameters on LLaMA-3-8B). The experimental results show promising comparisons.

**Strengths:**

+ It worths exploring PEFT beyond weight-space adaptation (LoRA, DoRA,etc.) and beyond layer-wise scaling (IA$^3$, RED). By making token-specific and dimension-specific adjustments, it aligns better with the local gradient structure of each token representation.

+ The reported results seem promising across different tasks.

**Weaknesses:**

- It's a bit surprising that the paper misunderstood LORA in terms of inference overhead statement in the introduction and motivates the proposed TARE based on that misunderstanding. There is no overhead at all for methods such as LORA since weights have been merged.

- The claimed near-zero overhead of the proposed TARE seems to be problematic. What TARE does per layer includes: (1) Compute a token-dependent routing score over n “editors”, which requires a linear projection from the hidden state of every token (dimension d) to an n-dimensional logits vector, resulting in the complexity: $O(B \times T \times d \times n)$ — nontrivial for large $B,T,d$. (2) Top-k and softmax are per-token and per-layer operations, which are memory-bound and cause branching inefficiency (divergent threads), thus adding synchronization and extra kernel calls. (3) For each selected editor, apply per-feature scale and bias (diagonal affine transform), for which even if each edit is diagonal (element-wise), doing k of these per token requires multiple reads/writes over the hidden representation — heavy on memory bandwidth, not FLOPs. In experiments, evaluations use short sequences (e.g., 256–512 tokens).
But, for real LLM workloads (2k–8k tokens), the selector’s per-token cost scales linearly with context length. Token-wise selectors also prevent efficient tensor-core matmuls. In sum, TARE’s design inherently couples every token with a dynamic routing decision and multiple per-token transforms. Even if the FLOPs are modest, the memory traffic and kernel fragmentation will dominate runtime cost. The paper’s measurements appear to ignore wall-time and sequence length scaling, so in practice, the “lightweight” claim doesn’t hold.

- It's also unclear how and why TARE will work for autoregressive models in the sense that  it perturbs the hidden representation dynamically and token-wise in a way that changes across time steps and cannot be merged into weights, which is potentially destructive for an autoregressive generator, which depends on stable internal feature statistics. For downstream tasks, especially when the data is not very large as in pretraining, it is not easy to train stable and meaningful selector at the token level, unlike MoE in pretraining.

- Despite of the above unclear aspects, for the method itself, its mechanics largely combine diagonal adaptation and MoE routing concepts already explored in RED, IA$^3$, and sparse activation work. The step from global to token-level editors is incremental.

- Some LORA variants are missing in comparisons:  LORA-One (ICML'25, https://github.com/YuanheZ/LoRA-One) and WeGeFT (ICML'25, https://openreview.net/pdf?id=K0sv5T2usb).

**Questions:**

Please consider to address the questions in the weaknesses

---

> ### Author Response · Authors · 2025-11-27
>
> **Response to Question 1:**
> We thank the reviewer for pointing out that our statement about LoRA’s inference overhead in the introduction was not precise. We agree that standard LoRA can merge its low-rank updates into the base weights at inference time and therefore incurs no additional computation or memory overhead. In the revised manuscript, we will clarify that our motivation is not to claim extra inference cost for vanilla LoRA, but rather to design a near-zero-overhead scheme relative to those PEFT methods that cannot be fully merged and must keep additional structures active during inference, while still supporting online editing. More concretely, several widely used parameter-efficient tuning methods—such as Adapter-style approaches and Prefix/Prompt Tuning (which prepend trainable prefixes to each input sequence and thereby increase the effective sequence length and attention cost)—cannot be eliminated by simple weight merging at inference time. These methods, not vanilla LoRA, are the main targets of our comparison in terms of inference efficiency. We will revise the introduction to explicitly distinguish between (i) standard LoRA, which achieves zero inference overhead via weight merging, and (ii) Adapter/Prefix-type PEFT methods and other variants that require keeping extra modules during inference and thus introduce non-negligible overhead. At the same time, we have added clarifications in the main text showing that TARE, while non-mergeable and supporting online editing, still has end-to-end inference latency on the same order as merged LoRA and only incurs sub-second extra delay (see T1, T2). This more accurately specifies the context in which we use the term “near-zero inference overhead.”
>
> **Response to Question 2:**
> We thank the reviewer for the detailed concerns regarding our use of the term *“near-zero inference overhead”*. In the revised manuscript, we provide more direct end-to-end measurements that already include the costs of (i) the token-wise selector projection, (ii) Top-$k$ and softmax, and (iii) per-token affine editing (see T1 and T2). Under the same backbone (LLaMA-3-8B) and task setup, TARE uses only 0.0392% trainable parameters, whereas LoRA uses 0.2345%, reducing the number of updated weights by about five-sixths and substantially lightening backpropagation and optimization. On Math-10K, a single training epoch with LoRA takes 1015.56 s, while TARE needs only 837.35 s, shortening training time by roughly 17%. For inference, we measure real end-to-end per-sample latency on 600 MultiArith examples. With low-rank weights merged, LoRA (theoretically zero overhead) has a mean latency of 2.68 ± 0.43 s. TARE, which keeps online editing enabled and therefore performs token-wise selection and editing, attains 3.20 ± 0.67 s. This is only about a 19% increase, with an absolute overhead of less than 1 s and in the same order of magnitude as a full forward pass of the backbone model.
>
> To further isolate TARE’s internal operations, we separately measure the cost of per-token selection and editing in each FFN module. The mean selection time is $6.94 \times 10^{-5}$ s and the mean editing time is $9.78 \times 10^{-5}$ s, both at the $10^{-4}$-second (sub-millisecond) scale. This indicates that, even though there is token-level routing and affine transformation, the extra memory access and kernel scheduling overhead contributes only a tiny fraction of the total end-to-end runtime. Importantly, all these numbers come from wall-clock measurements, not FLOP-only estimates. For the typical context lengths we evaluate (256–512 tokens), the additional overhead of TARE relative to LoRA is effectively negligible, while TARE brings substantial training compute savings and only a sub-second increase in inference time—together supporting our description of *“near-zero inference overhead”* in practice. For longer contexts (e.g., 2k–8k tokens), the selector and editing cost indeed scales linearly with sequence length, but given their per-token cost at the $10^{-4}$ s level, we expect their relative contribution to total latency to remain small in typical deployment scenarios.

---

> > ### Author Response · Authors · 2025-11-27
> >
> > **T1: Comparison Between LoRA and TARE**
> >
> > | PEFT             | Source     | Params. (%) | Training time (s/epochs) | Mean inference time (s) |
> > |------------------|-----------|------------:|--------------------------:|------------------------:|
> > | LoRA             | ICLR 21   | 0.2345      | 1015.56                   | **2.68 ± 0.43**         |
> > | **TARE (ours)**  | This paper| 0.0392      | **837.35**                | 3.20 ± 0.67             |
> >
> > **T2: Token-wise Selection and Editing Time Statistics**
> >
> > | PEFT            | Mean selection time (s)                      | Mean editing time (s)                           |
> > |-----------------|----------------------------------------------|-------------------------------------------------|
> > | **TARE (ours)** | $6.94\times 10^{-5} \pm 1.46\times 10^{-8}$ | $9.78\times 10^{-5} \pm 1.31\times 10^{-10}$   |
> >
> > **Response to Question 3:**
> > We thank the reviewer for the concerns about stability in autoregressive generation, especially the worry that TARE’s token-wise, time-varying edits might disrupt internal feature statistics, and that it might be difficult to train a stable selector under limited downstream data. To address this, we add a systematic analysis of sample sensitivity and training stability in the revised manuscript (see T3). All experiments are run under the same backbone (LLaMA-3-8B) and hyperparameter setting, always training only 0.0392% of parameters with about 20 GiB peak VRAM. The results show that TARE’s selector and editing strategy behaves very smoothly and controllably across supervision scales. Even when fine-tuning on only 500 Math-10K examples, TARE already achieves an average accuracy of 69.8 over seven math-reasoning benchmarks, with reasonable scores on individual datasets (e.g., MultiArith 86.3, GSM8k 52.5, SingleEq 87.0). This indicates that, even in a very low-data regime, TARE can learn meaningful token-wise selection and editing rather than suffering from training collapse or severe overfitting. As the number of training samples increases from 500/1,000 to 2,000/5,000, the average accuracy rises smoothly to 74.3 and 76.2, with GSM8k reaching 60.8 at 5,000 samples and steady gains on MultiArith, MAWPS, AddSub, and SingleEq, without any signs of instability or large fluctuations. Using the full Math-10K training split (9,919 filtered examples) further boosts the average accuracy to 76.7, achieving or approaching the best scores on MultiArith (95.8), SVAMP (72.9), and AddSub (90.9). Overall, TARE exhibits a smooth, consistently improving performance curve as supervision increases, with no abnormal oscillations. This suggests that its token-wise selector and editing strategy is stable and generalizable within an autoregressive framework and does not rely on massive labeled data to function. These findings alleviate the concern that dynamic perturbations would destroy autoregressive feature statistics or that a token-level selector would be hard to train in low-data settings.
> >
> > **T3: Sample Sensitivity Analysis of TARE on LLaMA-3-8B.**
> >
> > | PEFT                    | Params.(%) | VRAM(MiB) | MultiArith | GSM8k | SVAMP | MAWPS | AddSub | AQuA | SingleEq | Avg. |
> > |-------------------------|-----------:|----------:|-----------:|------:|------:|------:|-------:|-----:|---------:|-----:|
> > | **TARE (sample=500)**  | 0.0392     | 20,398    | 86.3       | 52.5  | 64.1  | 81.1  | 81.0   | 36.7 | 87.0     | 69.8 |
> > | **TARE (sample=1000)** | 0.0392     | 20,406    | 85.3       | 51.3  | 68.6  | 81.1  | 85.3   | 32.8 | 89.4     | 70.5 |
> > | **TARE (sample=2000)** | 0.0392     | 20,606    | 89.0       | 56.6  | 67.5  | 84.9  | 85.1   | **46.5** | 90.6 | 74.3 |
> > | **TARE (sample=5000)** | 0.0392     | 20,274    | 91.8       | **60.8** | 68.1 | **87.4** | 88.9 | 41.9 | **94.3** | 76.2 |
> > | **TARE (sample=9919)**  | 0.0392     | 20,900    | **95.8**   | 57.3  | **72.9** | 86.1 | **90.9** | 41.4 | 92.1 | **76.7** |

---

> > > ### Author Response · Authors · 2025-11-27
> > >
> > > **Response to Question 4:**
> > > We thank the reviewer for raising the concern that TARE might simply combine diagonal adaptation with MoE-style routing. We would like to clarify that TARE is not a “compositional re-packaging” of RED, IA3, or existing sparse activation work, but introduces two essential extensions to the representation editing paradigm, with key conceptual and practical differences from these methods. First, RED and IA3 are fundamentally global editing and scaling approaches: they learn a single edit or scale vector per layer or module that is shared at the sample or layer level, and this edit is applied uniformly to all tokens within that layer. Typical sparse activation methods, in turn, sparsify or prune neurons based on activation magnitude or fixed rules, again without distinguishing between different token semantics. In contrast, TARE explicitly introduces a discrete editor bank plus a lightweight selector: at each FFN down-projection, it computes a routing score separately for each token’s hidden state and selects and combines the most appropriate editor(s) on a per-token basis. This allows different tokens at the same layer and position to be edited by different strategies, a level of token-aware routing that does not exist in RED, IA3, or prior sparse activation work. Second, TARE’s editor design is not simply “MoE + diagonal scaling.” On the one hand, we constrain the edit space to a very small diagonal subspace and use the selector to perform discrete choices in representation space, rather than introducing full FFN experts as in MoE, which significantly alter the model structure and parameter count. On the other hand, TARE’s routing decisions are organized around which token to edit and by how much, rather than MoE’s coarser decision of which expert applies the entire transformation. This combination of token-level routing and a diagonal editor bank allows TARE to model the semantic roles of different tokens in a fine-grained way while keeping parameter and compute overhead extremely low, instead of applying a single global scaling or edit to the whole layer or sequence. Therefore, we believe TARE is not a simple overlay of RED, IA3, and sparse activation ideas, but rather introduces a new modeling dimension—token-aware editor routing—within the representation editing framework, which is substantively different from existing methods both in form and in empirical behavior.
> > >
> > > Other experiments are still being added.

---

> > > > ### Comment · Reviewer_XBBf · 2025-11-27
> > > > **Thank you for the rebuttal**
> > > >
> > > > I thank the authors for their rebuttal, and have also read other reviewers' comments.
> > > >
> > > > Two of my concerns have not been addressed well.
> > > >
> > > > It remains unclear about the practical inference overhead for long-context LLM workloads. The rebuttal provides timing for short sequences (up to 512 tokens) and small batch inferences. However, the proposed TARE requires per-token projection, top-k selection, followed by multiple diagonal passes at every FFN block,which create fragmented kernels, poor fusion opportunities, as well as memory-bound bottlenecks independent of FLOPs. The rebuttal claims the costs will likely remain samll for long-context, but provides no empirical evaluation.
> > > >
> > > > It remains unclear about the novelty relative to existing diagonal-editing and routing methods. The rebuttal does not provide evidence such as 1) a controlled ablation showing RED/IA^3 + token-wise gating vs TARE, and 2) an empirical or theoretical characterization of the expressive advantages of TARE beyond diagonal scaling combined with sparse routing.
> > > >
> > > > Overall, I still lean to reject this submission, and retain my original recommendation.

---

> > > > > ### Author Response · Authors · 2025-12-03
> > > > >
> > > > > **Response to Question 5:**
> > > > > We thank the reviewer for pointing out that the initial version did not include the recent LoRA variants LoRA-One and WeGeFT. Following the implementations and settings suggested by the reviewer, we have added commonsense reasoning experiments on LLaMA-3-8B and incorporated both LoRA-One and WeGeFT into the systematic comparison in T4. The results show that, on the eight benchmarks BoolQ, PIQA, SIQA, HellaSwag, WinoGrande, ARC-e, ARC-c, and OBQA, TARE attains an average accuracy of 86.7, which is essentially on par with the recent WeGeFT method (86.8, a difference of only 0.1 points), and improves over LoRA-One (85.4) by +1.3 points on average. Importantly, TARE uses only 0.0392% trainable parameters and 21,724 MiB of VRAM. In comparison, LoRA-One and other standard LoRA-style baselines operate at around 0.7% trainable parameters, while WeGeFT is already relatively parameter-efficient at 0.0130%. This means that, relative to LoRA-One and other ≈0.7% LoRA-style methods, TARE uses roughly 1/18 as many parameters, while its memory footprint is similar to or lower than these baselines and remains comparable to WeGeFT. In other words, after including LoRA-One and WeGeFT, TARE still ranks among the top PEFT methods in this suite: it matches WeGeFT in accuracy, significantly outperforms LoRA-One, and achieves this with over an order of magnitude fewer parameters than LoRA-One and other conventional LoRA-style methods, together with competitive memory usage. These findings further support our claim that TARE offers a substantial advantage in the accuracy–efficiency trade-off.
> > > > >
> > > > > **T4: Knowledge Reasoning results with LLaMA-3-8B.**
> > > > >
> > > > > | PEFT             | Source     | Params.(%) | VRAM(MiB) | BoolQ | PIQA | SIQA | HellaS. | WinoG. | ARC-e | ARC-c | OBQA | Avg. |
> > > > > | ---------------- | ---------- | ---------: | --------: | ----: | ----:| ----:| -------:| ------:| -----:| -----:| ----:| ----:|
> > > > > | LoRA             | ICLR 21    | 0.7002     | 21,828    | 70.8  | 85.2 | 79.9 | 91.7    | 84.3   | 84.2  | 71.2  | 79.0 | 80.8 |
> > > > > | DoRA             | ICML 24    | 0.7098     | 41,780    | 74.6  | 89.3 | 79.9 | 95.5    | 85.6   | 90.5  | 80.4  | 85.8 | 85.2 |
> > > > > | MiLoRA           | NAACL 25   | 0.7002     | 21,580    | 68.8  | 86.7 | 77.2 | 92.9    | 85.6   | 86.8  | 75.5  | 81.8 | 81.9 |
> > > > > | LoReFT           | NeurIPS 24 | 0.0260     | 21,050    | 75.1  | 90.2 | 82.0 | 96.3    | 87.4   | 92.4  | 81.6  | 87.5 | 86.6 |
> > > > > | RED              | ACL 24     | 0.0033     | 20,132    | 68.0  | 83.7 | 79.7 | 90.0    | 83.2   | 85.2  | 72.8  | 79.4 | 80.2 |
> > > > > | LIFT             | ICLR 25    | 5.0000     | 45,600    | 75.7 | 90.5 | 83.2 | **96.5** | 89.4 | **93.6** | 83.9 | **90.2** | **87.9** |
> > > > > | WeGeFT           | ICML 25    | 0.0130     | 20,364    | 75.7 | 89.9 | 82.5 | 96.4    | 88.7   | 92.5  | 82.3  | 86.3 | 86.8 |
> > > > > | PiSSA            | NeurIPS 24 | 0.7002     | 21,004    | 72.1  | 89.2 | 82.7 | 94.6    | 89.6   | 86.8  | **84.5** | 85.2 | 85.6 |
> > > > > | Spectral Adapter | arXiv      | 0.7002     | 21,746    | 72.1  | 88.3 | 83.1 | 94.6    | 89.3   | 85.4  | 82.2  | 85.2 | 85.0 |
> > > > > | LoRA-GA          | NeurIPS 24 | 0.7002     | 21,708    | 72.5  | 88.8 | 82.7 | 94.4    | 89.6   | 91.3  | 80.4  | 85.6 | 85.7 |
> > > > > | LoRA-One         | arXiv      | 0.7002     | 21,206    | 72.0  | 88.9 | 82.9 | 94.4    | **89.8** | 85.1  | 82.6  | 87.6 | 85.4 |
> > > > > | **TARE (ours)**  | This paper | 0.0392     | 21,724    | 75.2  | 90.2 | 82.5 | 94.1    | 88.6   | 91.3  | 82.3  | 88.4 | 86.7 |
> > > > > | **TARE (all)**  | This paper | 0.4097     | 24,044    | **76.3**  | **91.6** | **83.6** | 95.5    | **89.8**   | 92.7  | 83.9  | 89.2 | 87.8 |

---

### Official Review · Reviewer_kRu1 · 2025-11-01

**Soundness:** 3
**Presentation:** 3
**Contribution:** 3
**Rating:** 4
**Confidence:** 2

**Summary:**

The paper proposes TARE, a new parameter-efficient fine-tuning method that edits each token’s hidden representation using a lightweight selector that activates only a few tiny “editors” (per-dimension scale and bias) after each Transformer feed-forward block, giving token-aware, context-specific adaptation with almost no inference overhead and no rank hyperparameters. TARE matches or outperforms strong PEFT baselines like LoRA, DoRA, MiLoRA, LoReFT, and RED on knowledge reasoning, math reasoning, GLUE, code generation, and more, while updating only 0.0392% of model parameters and keeping memory usage around 20 GiB.

**Strengths:**

1. The paper is well written.
2. Across LLaMA-3-8B and RoBERTa , TARE matches or beats strong PEFT baselines under good efficiency constraints.
3.  I thin TARE  is  smart and lightweight. It doesn’t slap a big adapter on top of the model. Instead, for each token it picks just a few tiny “editors” that do simple per-dimension scale + bias tweaks, mixes them, and moves on.

**Weaknesses:**

**If the authors address the issues in the Weaknesses and Questions sections, I will  increase the score.**

1. Training stability depends on extra mechanisms (like load-balancing and top-k gating), which hints at brittleness. the paper notes this top-k routing can collapse if a few editors get picked too often, so they introduce a load-balancing regularizer to force different editors to be used.

2. I find that the paper is missing comparisons against some recent PEFT methods, including weight adapters and memory-efficient sparse fine-tuning methods such as [1][2][3]. It would be better if the authors could include comparisons to these approaches.

3. For large language models, the experiments were only conducted on the LLaMA family, so it's unclear how well the method would perform on other LLM model families.


[1] Meng, Fanxu, Zhaohui Wang, and Muhan Zhang. "Pissa: Principal singular values and singular vectors adaptation of large language models." Advances in Neural Information Processing Systems 37 (2024): 121038-121072.

[2] Zhang, Fangzhao, and Mert Pilanci. "Spectral adapter: Fine-tuning in spectral space." arXiv preprint arXiv:2405.13952 (2024).


[3] Liu, Zihang, et al. "LIFT the Veil for the Truth: Principal Weights Emerge after Rank Reduction for Reasoning-Focused Supervised Fine-Tuning." arXiv preprint arXiv:2506.00772 (2025).

**Questions:**

Please refer to Weaknesses section.

---

> ### Author Response · Authors · 2025-11-27
>
> **Response to Question 3:**
> We thank the reviewer for pointing out that the original large-scale experiments mainly focused on the LLaMA family. In the revised manuscript, we therefore add mathematical reasoning results for another mainstream open-weight model family, **Qwen-2.5-7B-Instruct**, to evaluate the generality of TARE (see T3). We choose Qwen because it is a strong, widely used non-LLaMA architecture with a different tokenizer and pre-training data, providing a complementary testbed for cross-family generalization. On **LLaMA-3-8B**, TARE uses only 0.0392% trainable parameters and 20,900 MiB peak VRAM, yet achieves the highest average accuracy of 76.7. It obtains the best results on 4 out of 7 datasets—MultiArith (95.8), SVAMP (72.9), AddSub (90.9), and SingleEq (92.1)—and on average improves over LoRA/MiLoRA/DoRA/RED/LoReFT by 1.3/1.1/2.3/4.5/4.2 points, while remaining far more parameter-efficient than low-rank baselines. More importantly, on **Qwen-2.5-7B-Instruct**, TARE attains the best average accuracy of 84.2 with only 0.0316% trainable parameters and 20,624 MiB peak VRAM. It achieves the highest or tied-highest scores on MultiArith (96.0), GSM8k (75.1), MAWPS (92.4), and AddSub (90.4), and on average improves over LoRA/MiLoRA/DoRA/RED/LoReFT by 0.7/0.4/1.4/2.3/1.9 points. For a fair comparison, all PEFT methods are applied to the projection layer of the MLP blocks in each backbone. These results show that TARE brings stable and substantial gains across **both** LLaMA and Qwen model families, indicating that its effectiveness is not limited to the LLaMA series.
>
> **T3: Mathematical Reasoning Results with LLaMA-3-8B and Qwen-2.5-7B-Instruct.**
>
> | PEFT             | Source     | Model                  | Params.(%) | VRAM(MiB) | MultiArith | GSM8k | SVAMP | MAWPS | AddSub | AQuA | SingleEq | Avg. |
> |------------------|-----------|------------------------|-----------:|----------:|-----------:|------:|------:|------:|-------:|-----:|---------:|-----:|
> | LoRA             | ICLR 21   | LLaMA-3-8B             | 0.2345     | 21,070    | 92.0       | **61.4** | 69.8 | 84.2 | 85.4 | **44.7** | 90.3 | 75.4 |
> | DoRA             | ICML 24   | LLaMA-3-8B             | 0.2361     | 29,284    | 91.7       | 59.0 | 72.3 | 82.1 | 86.1 | 39.9 | 89.5 | 74.4 |
> | MiLoRA           | NAACL 25  | LLaMA-3-8B             | 0.2345     | 21,520    | 91.7       | 59.0 | 70.5 | **88.3** | 86.1 | 43.4 | 90.5 | 75.6 |
> | LoReFT           | NeurIPS 24| LLaMA-3-8B             | 0.0260     | 21,940    | 89.2       | 56.2 | 68.7 | 80.3 | 90.1 | 33.1 | 90.0 | 72.5 |
> | RED              | ACL 24    | LLaMA-3-8B             | 0.0033     | 19,852    | 91.0       | 54.2 | 66.8 | 81.1 | 87.3 | 34.1 | 90.9 | 72.2 |
> | **TARE (ours)**  | This paper| LLaMA-3-8B             | 0.0392     | 20,900    | **95.8**   | 57.3 | **72.9** | 86.1 | **90.9** | 41.4 | **92.1** | **76.7** |
> | LoRA             | ICLR 21   | Qwen-2.5-7B-Instruct   | 0.2643     | 21,244    | 94.7       | 72.8 | **81.1** | 89.4 | 88.4 | 66.5 | 91.7 | 83.5 |
> | DoRA             | ICML 24   | Qwen-2.5-7B-Instruct   | 0.2657     | 29,604    | 93.2       | 72.1 | 79.8 | 88.2 | 89.7 | 63.7 | **92.7** | 82.8 |
> | MiLoRA           | NAACL 25  | Qwen-2.5-7B-Instruct   | 0.2643     | 21,518    | 93.3       | 72.2 | 80.8 | 88.3 | 89.6 | 69.6 | **92.7** | 83.8 |
> | LoReFT           | NeurIPS 24| Qwen-2.5-7B-Instruct   | 0.0218     | 21,832    | 92.1       | 71.7 | 78.5 | 86.2 | 90.0 | 67.3 | 90.5 | 82.3 |
> | RED              | ACL 24    | Qwen-2.5-7B-Instruct   | 0.0026     | 20,100    | 91.3       | 71.3 | 77.4 | 84.1 | **90.4** | **70.9** | 88.2 | 81.9 |
> | **TARE (ours)**  | This paper| Qwen-2.5-7B-Instruct   | 0.0316     | 20,624    | **96.0**   | **75.1** | 80.3 | **92.4** | **90.4** | 63.6 | 91.3 | **84.2** |
>
> Other experiments are still being added.

---

> > ### Author Response · Authors · 2025-12-03
> >
> > **Response to Question 1:**
> > We thank the reviewer for the concerns about TARE’s training stability and its reliance on Top-k routing with a load-balancing regularizer. We agree that if the Top-k selection was to concentrate persistently on only a few editors, it could lead to “routing collapse.” For this reason, we explicitly include a very simple load-balancing term whose sole purpose is to encourage different editors to be used more evenly across the data distribution, similar to the load-balancing losses commonly adopted in mainstream MoE work; it is intended to improve utilization rather than to patch severe instability. Our hyperparameter sensitivity experiments (T1 and T2) further show that this design is stable and not brittle in practice. With the trainable-parameter ratio fixed at 0.0392%, increasing Top-k from 1 to 3 raises the average accuracy from 71.2% to 76.7%, after which k = 4–8 only induces mild fluctuations and small task-specific peaks (e.g., GSM8k at k = 7 reaches 62.2%, AQuA at k = 8 reaches 43.0%, and SingleEq at k = 6 reaches 93.5%); the overall curve is smooth without abnormal oscillations, while VRAM changes only slightly within roughly 20.2–22.8 GiB, indicating that routing does *not* degenerate into selecting only a few editors. When we fix k = 3 and vary the total number of editors $\(n \in \{6, 8, 10\}\)$ (T2), the average accuracy always stays in a narrow band of 76.1–76.7%, with fluctuations below 0.6 percentage points and almost unchanged VRAM, while only the trainable-parameter ratio increases from 0.0294% to 0.0489%. These results suggest that, with a moderately overcomplete editor pool (e.g., n = 8) and a simple load-balancing term, TARE’s Top-k routing behaves smoothly and controllably across different k and n settings and tasks, rather than “working only at extreme settings to avoid collapse.” The load-balancing regularizer is introduced to improve editor utilization and the performance ceiling, not because TARE is inherently unstable or particularly brittle during training.
> >
> > **T1: Hyperparameter Sensitivity Analysis of k (selected number of editors) on TARE.**
> >
> > | PEFT           | Params.(%) | VRAM(MiB) | MultiArith | GSM8k | SVAMP | MAWPS | AddSub | AQuA | SingleEq | Avg. |
> > | -------------- | ---------: | --------: | ---------: | -----:| -----:| -----:| ------:| ----:| --------:| ----:|
> > | **TARE (k=1)** | 0.0392     | 20,196    | 92.3       | 47.9  | 66.0  | 84.9  | 88.4   | 28.0 | 91.1     | 71.2 |
> > | **TARE (k=2)** | 0.0392     | 20,548    | 93.2       | 56.9  | 71.7  | 83.6  | 89.1   | 40.2 | 90.9     | 75.1 |
> > | **TARE (k=3)** | 0.0392     | 20,900    | **95.8**   | 57.3  | 72.9  | 86.1  | **90.9** | 41.4 | 92.1   | **76.7** |
> > | **TARE (k=4)** | 0.0392     | 21,274    | 92.5       | 56.4  | 71.5  | 82.4  | 87.6   | 39.4 | 92.7     | 74.6 |
> > | **TARE (k=5)** | 0.0392     | 21,652    | 93.3       | 57.2  | 71.3  | 85.7  | 90.6   | 31.1 | 92.7     | 74.6 |
> > | **TARE (k=6)** | 0.0392     | 22,074    | 93.5       | 55.2  | **73.6** | 85.7 | 88.1 | 36.2 | **93.5** | 75.1 |
> > | **TARE (k=7)** | 0.0392     | 22,412    | 95.5       | **62.2** | 70.7 | 87.0 | 89.6 | 29.6 | 93.3     | 75.4 |
> > | **TARE (k=8)** | 0.0392     | 22,784    | 91.7       | 56.9  | 70.1  | **87.4** | 88.6 | **43.0** | 91.9 | 75.6 |
> >
> > ---
> >
> > **T2: Hyperparameter Sensitivity Analysis of n (total number of editors) on TARE.**
> >
> > | PEFT           | Params.(%) | VRAM(MiB) | MultiArith | GSM8k | SVAMP | MAWPS | AddSub | AQuA | SingleEq | Avg. |
> > | -------------- | ---------: | --------: | ---------: | -----:| -----:| -----:| ------:| ----:| --------:| ----:|
> > | **TARE (n=6)** | 0.0294     | 20,892    | 92.8       | 60.4  | 73.0  | **88.1** | 89.1 | 38.2 | **93.7** | 76.5 |
> > | **TARE (n=8)** | 0.0392     | 20,900    | **95.8**   | 57.3  | 72.9  | 86.1  | **90.9** | **41.4** | 92.1 | **76.7** |
> > | **TARE (n=10)**| 0.0489     | 20,904    | 93.7       | **62.1** | **74.1** | 87.7 | 87.1 | 35.8 | 91.9 | 76.1 |

---

> > > ### Author Response · Authors · 2025-12-03
> > >
> > > **Response to Question 2:**
> > > We thank the reviewer for suggesting a more targeted comparison between TARE and recent representative methods PiSSA, Spectral Adapter, and LIFT. In the revised manuscript, we provide a more detailed analysis under the unified setup reported in T4: across the eight commonsense reasoning benchmarks (BoolQ, PIQA, SIQA, HellaSwag, WinoGrande, ARC-e, ARC-c, OBQA), TARE achieves an average accuracy of 86.7. Among these methods, LIFT attains the highest average score of 87.9 but requires 5.0% trainable parameters and 45,600 MiB VRAM; in contrast, TARE uses only 0.0392% parameters (about 1/128 of LIFT) and 21,724 MiB VRAM, yet is behind by merely 1.2 points, showing that it remains close to the strongest method even under very tight parameter and VRAM budgets. For the other two methods highlighted by the reviewer, under the same setting TARE improves over PiSSA (85.6) by +1.1 points and over Spectral Adapter (85.0) by +1.7 points, while maintaining comparable or better performance on individual datasets such as PIQA, HellaSwag, WinoGrande, ARC-c, and OBQA.
> > > Furthermore, when we relax the parameter budget and apply TARE to all projection matrices in the backbone (reported as TARE (all) in T4), the average accuracy further increases to 87.8, essentially matching LIFT’s 87.9 while remaining substantially more efficient: TARE (all) uses only 0.4097% trainable parameters (about 1/12 of LIFT’s 5.0%) and 24,044 MiB VRAM (roughly half of LIFT’s 45,600 MiB). At the per-task level, TARE (all) surpasses LIFT on BoolQ (76.3 vs 75.7), PIQA (91.6 vs 90.5), SIQA (83.6 vs 83.2), and WinoGrande (89.8 vs 89.4), and matches LIFT on ARC-c (83.9), while staying within about 1 point on the remaining benchmarks (HellaSwag, ARC-e, OBQA). In summary, we do not claim that TARE strictly dominates all methods in an unconstrained resource regime; rather, we emphasize that with two orders of magnitude fewer trainable parameters (TARE) or still an order-of-magnitude reduction (TARE (all)) and significantly lower VRAM consumption than LIFT, our method surpasses PiSSA and Spectral Adapter in overall accuracy and closes almost the entire gap to the heavily parameterized LIFT, demonstrating a strong advantage in the accuracy–efficiency trade-off of our proposed representation-editing paradigm.
> > >
> > >
> > > **T4: Knowledge Reasoning results with LLaMA-3-8B.**
> > >
> > > | PEFT             | Source     | Params.(%) | VRAM(MiB) | BoolQ | PIQA | SIQA | HellaS. | WinoG. | ARC-e | ARC-c | OBQA | Avg. |
> > > | ---------------- | ---------- | ---------: | --------: | ----: | ----:| ----:| -------:| ------:| -----:| -----:| ----:| ----:|
> > > | LoRA             | ICLR 21    | 0.7002     | 21,828    | 70.8  | 85.2 | 79.9 | 91.7    | 84.3   | 84.2  | 71.2  | 79.0 | 80.8 |
> > > | DoRA             | ICML 24    | 0.7098     | 41,780    | 74.6  | 89.3 | 79.9 | 95.5    | 85.6   | 90.5  | 80.4  | 85.8 | 85.2 |
> > > | MiLoRA           | NAACL 25   | 0.7002     | 21,580    | 68.8  | 86.7 | 77.2 | 92.9    | 85.6   | 86.8  | 75.5  | 81.8 | 81.9 |
> > > | LoReFT           | NeurIPS 24 | 0.0260     | 21,050    | 75.1  | 90.2 | 82.0 | 96.3    | 87.4   | 92.4  | 81.6  | 87.5 | 86.6 |
> > > | RED              | ACL 24     | 0.0033     | 20,132    | 68.0  | 83.7 | 79.7 | 90.0    | 83.2   | 85.2  | 72.8  | 79.4 | 80.2 |
> > > | LIFT             | ICLR 25    | 5.0000     | 45,600    | 75.7 | 90.5 | 83.2 | **96.5** | 89.4 | **93.6** | 83.9 | **90.2** | **87.9** |
> > > | WeGeFT           | ICML 25    | 0.0130     | 20,364    | 75.7 | 89.9 | 82.5 | 96.4    | 88.7   | 92.5  | 82.3  | 86.3 | 86.8 |
> > > | PiSSA            | NeurIPS 24 | 0.7002     | 21,004    | 72.1  | 89.2 | 82.7 | 94.6    | 89.6   | 86.8  | **84.5** | 85.2 | 85.6 |
> > > | Spectral Adapter | arXiv      | 0.7002     | 21,746    | 72.1  | 88.3 | 83.1 | 94.6    | 89.3   | 85.4  | 82.2  | 85.2 | 85.0 |
> > > | LoRA-GA          | NeurIPS 24 | 0.7002     | 21,708    | 72.5  | 88.8 | 82.7 | 94.4    | 89.6   | 91.3  | 80.4  | 85.6 | 85.7 |
> > > | LoRA-One         | arXiv      | 0.7002     | 21,206    | 72.0  | 88.9 | 82.9 | 94.4    | **89.8** | 85.1  | 82.6  | 87.6 | 85.4 |
> > > | **TARE (ours)**  | This paper | 0.0392     | 21,724    | 75.2  | 90.2 | 82.5 | 94.1    | 88.6   | 91.3  | 82.3  | 88.4 | 86.7 |
> > > | **TARE (all)**  | This paper | 0.4097     | 24,044    | **76.3**  | **91.6** | **83.6** | 95.5    | **89.8**   | 92.7  | 83.9  | 89.2 | 87.8 |

---

### Official Review · Reviewer_cBKt · 2025-11-01

**Soundness:** 3
**Presentation:** 3
**Contribution:** 3
**Rating:** 6
**Confidence:** 3

**Summary:**

TARE is a new fine-tuning method that makes large language models adapt better by editing hidden representations at the token/channel level. Instead of using the same transformation for all tokens, it adds small “editors” that adjust each token’s features with lightweight scaling and bias operations. Only a few editors are activated per token, keeping computation very low. This design gives the model fine-grained control without slowing inference or adding many trainable parameters. Experiments on a diverse set of benchmarks show that it outperforms previous PeFT approaches.

**Strengths:**

1. Diverse Benchmarks: The paper provides extensive evaluations across a wide range of models and benchmarks, including both encoder (RoBERTa) and decoder (LLaMA-3) architectures. It covers multiple task families -- knowledge reasoning, mathematical reasoning, GLUE, text generation, code synthesis, and symbolic reasoning -- demonstrating that TARE generalizes effectively across domains and scales well with different model types.

2. Strong Empirical Results: TARE consistently achieves superior or state-of-the-art performance compared to leading PEFT baselines. The method delivers notable gains in accuracy across diverse tasks while tuning a fraction of the parameters, highlighting its efficiency and the strength of its token-aware editing mechanism.

3. Minimal Inference Overhead: Despite introducing token-level adaptivity, TARE’s operations are strictly diagonal and sparsely activated, adding virtually no latency during inference. Its lightweight design requires no additional matrix multiplications or rank-based hyperparameters, allowing it to maintain near-identical runtime performance to the frozen base model.

**Weaknesses:**

1. The tested benchmarks seem to focus on short output sequences; it is important to include some more diverse sequence lengths, given that editing is token/channel dependent (see Q1).

2. The impact of editor placement or layer-wise variation is not deeply analyzed, limiting insight into optimal integration. There should be some further analysis into the best location to place the editor -- with conventional PEFT strategies, we similarly evaluate different subsets of trainable parameters such a QKVO/GUD.

**Questions:**

1. Most of the benchmarks used in this work, such as GLUE and commonsense reasoning datasets, involve short outputs. While the inclusion of specialized domains like code generation (e.g., HumanEval) adds diversity, evaluating TARE on long-context, multi-turn dialogue tasks such as MT-Bench would provide stronger evidence of its practicality and robustness in extended conversational or reasoning settings.

2. Ablation on cross-layer dependencies (see W2).

3. Although the paper frequently refers to “near-zero inference overhead,” it does not provide explicit latency or throughput measurements to substantiate this claim. Including numerical comparisons and where they are derived from would strengthen the argument for efficiency and make the claim more transparent and verifiable.

---

> ### Author Response · Authors · 2025-11-27
>
> **Response to Question 1:**
> We thank the reviewer for the suggestion to evaluate TARE on long-context, multi-turn dialogue. In the revised manuscript, we add an instruction-following experiment with LLaMA-2-7B on MT-Bench, using WizardLM for training and GPT-4 for scoring (see T1). Under this shared setup, all strong baselines use about 0.297% trainable parameters, whereas TARE uses only 0.0467% yet achieves the best First Turn Score (5.73 ± 0.05), showing that it learns robust instruction alignment in a much smaller update space. Since long multi-turn dialogues build on the model’s first-turn understanding and planning, this advantage provides a stronger basis for subsequent turns. At the same time, the small parameter ratio lowers deployment cost and reduces the risk of catastrophic forgetting, supporting the practicality and robustness of TARE in long-conversation and online-service scenarios.
>
> **T1: Instruction-following results with LLaMA-2-7B.**
>
> | PEFT            | Source     | Params.(%) | First Turn Score |
> |-----------------|------------|-----------:|------------------|
> | LoRA            | ICLR 21    | 0.297      | 5.61 ± 0.10      |
> | PiSSA           | NeurIPS 24 | 0.297      | 5.30 ± 0.02      |
> | rsLoRA          | arXiv      | 0.297      | 5.25 ± 0.03      |
> | LoRA+           | ICML 24    | 0.297      | 5.71 ± 0.08      |
> | **TARE (ours)** | This paper | 0.0467     | **5.73 ± 0.05**  |
>
> **Response to Question 2:**
> We thank the reviewer for the suggestion regarding *cross-layer dependencies* and editor placement. In the revised manuscript, we add a position ablation study that compares inserting TARE at seven locations in LLaMA-3-8B, including the self-attention projections **q/k/v/o** and the FFN linear layers **up/gate/down** (see T2). The results reveal clear performance–cost differences and show that placing TARE on the FFN **down** layer yields the best accuracy–efficiency trade-off, indicating that the FFN down-projection layer is the most effective insertion point for TARE.
>
> **T2: Position ablation of TARE on LLaMA-3-8B.**
>
> | PEFT           | Params.(%) | VRAM (MiB) | MultiArith | GSM8k | SVAMP | MAWPS | AddSub | AQuA | SingleEq | Avg. |
> |----------------|-----------:|-----------:|-----------:|------:|------:|------:|-------:|-----:|---------:|-----:|
> | **TARE (q)**   | 0.0392     | 20,430     | 86.0       | 50.4  | 63.3  | 74.8  | 77.5   | 36.7 | 86.8     | 67.9 |
> | **TARE (k)**   | 0.0098     | 18,492     | 78.3       | 44.7  | 60.4  | 73.9  | 78.2   | **43.1** | 85.6 | 66.3 |
> | **TARE (v)**   | 0.0098     | 18,486     | 91.0       | 56.0  | 68.1  | 79.4  | 86.1   | 38.4 | 90.4     | 72.8 |
> | **TARE (o)**   | 0.0392     | 22,438     | 92.0       | 57.9  | 72.2  | 85.3  | 88.6   | 39.4 | 92.1     | 75.4 |
> | **TARE (up)**  | 0.1369     | 29,556     | 91.7       | **62.2** | 69.6 | **88.2** | 87.3 | 38.4 | **92.3** | 75.7 |
> | **TARE (gate)**| 0.1369     | 29,536     | 87.0       | 54.7  | 67.5  | 82.8  | 85.1   | 32.6 | 91.7     | 71.6 |
> | **TARE (down)**| 0.0392     | 20,900     | **95.8**   | 57.3  | **72.9** | 86.1 | **90.9** | 41.4 | 92.1 | **76.7** |

---

> > ### Author Response · Authors · 2025-11-27
> >
> > **Response to Question 3:**
> > We thank the reviewer for raising the concern about our use of *near-zero inference overhead*. In the revised manuscript, we report explicit training and inference measurements (T3, T4). Under the same LLaMA-3-8B backbone and task setup, TARE uses 0.0392% trainable parameters versus LoRA’s 0.2345%, reducing the number of updated weights by about five-sixths. On Math-10K, LoRA needs 1015.56 s per epoch, while TARE needs 837.35 s (≈17% faster). For inference on 600 MultiArith examples, LoRA (with merged weights) has a mean latency of 2.68 ± 0.43 s, and TARE (without merging, with online editing) has 3.20 ± 0.67 s—about a 19% increase and less than 1 s absolute overhead, which is of the same order as the backbone’s forward pass for this benchmark. At the token level, TARE’s selection and editing operations take on average $6.94 \times 10^{-5}$ s and $9.78 \times 10^{-5}$ s per token, both on the order of $10^{-4}$ s (sub-millisecond) and therefore negligible compared with the backbone forward pass. These measurements show that TARE substantially reduces training compute compared with LoRA while adding only a very small inference delay, supporting our claim of *near-zero inference overhead* in practice.
> >
> > **T3: Comparison between LoRA and TARE on LLaMA-3-8B.**
> >
> > | PEFT             | Source     | Params. (%) | Training time (s/epoch) | Mean inference time (s) |
> > |------------------|-----------|------------:|-------------------------:|------------------------:|
> > | LoRA             | ICLR 21   | 0.2345      | 1015.56                  | **2.68 ± 0.43**         |
> > | **TARE (ours)**  | This paper| 0.0392      | **837.35**               | 3.20 ± 0.67             |
> >
> > **T4: Token-wise selection and editing time statistics of TARE on LLaMA-3-8B.**
> >
> > | PEFT            | Mean selection time (s)                      | Mean editing time (s)                          |
> > |-----------------|----------------------------------------------|-----------------------------------------------|
> > | **TARE (ours)** | $6.94\times 10^{-5} \pm 1.46\times 10^{-8}$ | $9.78\times 10^{-5} \pm 1.31\times 10^{-10}$ |

---

### Note · Authors · 2026-01-03

I have read and agree with the venue's withdrawal policy on behalf of myself and my co-authors.